**Article** https://doi.org/10.1038/s41467-024-44984-y

# Rapid and visual identification of *β*-lactamase subtypes for precision antibiotic therapy

Wenshuai Li[1,2], Jingqi Li[1,2], Hua Xu[3], Hongmei Gao[3] & Dingbin Liu [1,2] ✉

The abuse of antibiotics urgently requires rapid identification of drug-resistant bacteria at the point of care (POC). Here we report a visual paper sensor that allows rapid (0.25-3 h) discrimination of the subtypes of *β*-lactamase (the major cause of bacterial resistance) for precision antibiotic therapy. The sensor exhibits high performance in identifying antibiotic-resistant bacteria with 100 real samples from patients with diverse bacterial infections, demonstrating 100% clinical sensitivity and specificity. Further, this sensor can enhance the accuracy of antibiotic use from 48% empirically to 83%, and further from 50.6% to 97.6% after eliminating fungal infection cases. Our work provides a POC testing platform for guiding effective management of bacterial infections in both hospital and community settings.

Antibiotic-resistant bacteria are the most commonly identified pathogens that often cause severe infections in both hospital and community settings[1–3]. As such, timely and appropriate antibiotic therapy administration is crucial for managing bacterial infections in the clinic[4]. However, patients are often treated with broad-spectrum antimicrobial drugs due to the lack of rapid identification methods for bacteria. Unfortunately, there is a frequent mismatch between infection severity and antibiotic prescription, leading to the overuse of antibiotics or/and inappropriate drug selection[5–7]. Moreover, exposure to broad-spectrum antibiotics may contribute to the emergence of "superbugs", thereby resulting in therapeutic failure[8]. Effective antimicrobial drug management calls for reliable identification of antibiotic resistance in bacterial infections.

Antimicrobial susceptibility testing (AST) is the gold-standard method to guide the use of antimicrobial drugs[9]. However, AST relies on bacterial culture, which often takes 1-7 days, and thus physicians have to empirically prescribe broad-spectrum antibiotics to treat bacterial infections before obtaining the testing results (Fig. 1A). In clinical practice, the accuracy of empirical prescription of antibiotics is less than 50%[10]. Moreover, AST may occasionally produce false-negative results because some bacteria are difficult to culture. Diverse alternative methods have emerged to expedite the

diagnostic process, encompassing RT-PCR[11,12], gene sequencing[13,14], next-generation sequencing[15,16], mass spectrometry[17,18], optical detection[19–26], thermal measurement[27], nanomechanical sensing[28], and plasmonic nanosensors[29]. Nevertheless, these methods usually require expensive equipment, skilled personnel, and complex sample processing, thus limiting their use in laboratory settings. It is urgently needed to develop approaches that are sensitive, rapid, and straightforward for point-of-care testing (POCT) of bacterial resistance.

In the clinic, the second- and third-generation cephalosporins are commonly prescribed as the primary treatment for bacterial infections[1,10]. Unfortunately, the emergence and dissemination of *β*-lactamases have significantly reduced the efficacy of these antibiotics. Based on the spectrum of *β*-lactam hydrolysis, *β*-lactamases can be classified into four major groups: broad-spectrum *β*-lactamase (BSBL), extended-spectrum *β*-lactamase (ESBL), AmpC *β*-lactamase (AmpC), and carbapenemase[30,31]. Of particular concern, carbapenem antibiotics are the last resort in combating life-threatening infections[32,33]. The treatment regimen to combat antibiotic resistance is made according to the *β*-lactamase subtypes in bacteria strains. Hence, the accurate identification of *β*-lactamase subtypes could enable tailored antibiotic treatment regimens that align with the specific infection patterns

[1]State Key Laboratory of Medicinal Chemical Biology, Frontiers Science Centers for Cell Responses and New Organic Matter, College of Chemistry, Nankai University, Tianjin 300071, China. [2]Tianjin Key Laboratory of Molecular Recognition and Biosensing, Nankai University, Tianjin 300071, China. [3]Department of Intensive Care Unit, Key Laboratory for Critical Care Medicine of the Ministry of Health, Emergency Medicine Research Institute, Tianjin First Center Hospital, School of Medicine, Nankai University, Tianjin 300071, China. ✉e-mail: liudb@nankai.edu.cn

exhibited by individual patients, thus reducing the abuse of antibiotics and ultimately improving patient outcomes.

Here, we report a POCT diagnostic sensor for rapid and visual identification of $\beta$-lactamase subtypes in diverse clinical samples. We show the sensor is able to discriminate the $\beta$-lactamase-producing bacteria in 100 real clinical samples with 100% accuracy within 0.25–3 h, enhancing the accuracy of antibiotic therapy from 48% to 83%, and further from 50.6% to 97.6% after eliminating fungal interference. We previously showed that the conjugation of -N$^+$(CH$_3$)$_3$ to a chromogenic carbapenem substrate (**CCS**) could dramatically improve the sensitivity and specificity of the probe in detecting carbapenemases[34]. Inspired by this, here we design three -N$^+$(CH$_3$)$_3$-bearing chromogenic cephalosporin substrates (**CCepS-N$^+$(CH$_3$)$_3$**) to detect the remaining three $\beta$-lactamase subtypes, including BSBL, ESBL, and AmpC, incorporating with **CCS-N$^+$(CH$_3$)$_3$** on a sensor to achieve the visual classification of $\beta$-lactamase-associated bacterial resistance. This $\beta$-lactamase subtype visualization (BSV) sensor can detect clinical bacteria-resistant isolates with the lowest detectable concentration at 10$^4$ CFU/mL, 1–2 orders of magnitude lower than other visual $\beta$-lactamase probes[35]. The BSV sensor can provide results within 0.25–3 h, depending on the types of biofluids being screened, which is much faster than methods relying on cultivation (Fig. 1A), holding great promise for the timely administration of antibiotics.

## Results

### Rational design of the BSV sensor and synthesis of CCepSs

The BSV sensor is a transparent polyethylene device with six independent channels where the probe-loaded paper sheets are installed in the chambers. The paper sensor includes a negative control in chamber 1, which maintains a stable lemon-yellow color to ensure the stability of the sensor quality. Chamber 2 is loaded with bromophenol blue to act as a positive control, which undergoes a notable color change from yellow to blue upon binding to proteins in the samples. Detection chambers 3–6 were assigned to the molecular probes targeting BSBL, ESBL, AmpC, and carbapenemase sequentially, based on their $\beta$-lactamase hydrolysis ability (Fig. 1B). The samples (20 µL) are dropped into the central sampling hole and then quickly diffuse into the chambers via capillary siphoning. The result can be visually identified by the paper sheets after a short incubation.

As previously reported[34], when the $\beta$-lactam ring of **CCS-N$^+$(CH$_3$)$_3$** is hydrolyzed by carbapenemase, the nitrogen atom in the amide converts into a secondary amine, resulting in a resonance change that elongates the conjugated system, thus leading to the redshift of absorption bands in the visible spectrum, accompanied by a significant color change from yellow to red. In this study, we wanted to expand the color-changing mechanism of **CCS-N$^+$(CH$_3$)$_3$** to the **CCepS** probes that can report the presence of BSBL, ESBL, and AmpC, respectively. The core structure of cephalosporins is 7-aminocephalosporanic acid (7-ACA)[36,37], whose 3-position was conjugated with a -N$^+$(CH$_3$)$_3$-bearing benzene derivative via a double bond to produce **CCepS-N$^+$(CH$_3$)$_3$–1**, which can be hydrolyzed by all kinds of $\beta$-lactamases (Fig. 1C). When the C7-$\beta$ position of **CCepS-N$^+$(CH$_3$)$_3$–1** was replaced with (E)−1-(2-aminothiazol-4-yl)ethan-1-one oxime, the resulting probe (namely **CCepS-N$^+$(CH$_3$)$_3$–2**) can resist the hydrolysis of BSBL[38,39], but can still be hydrolyzed by ESBL, AmpC, and carbapenemase. Further, when modified the C7-$\alpha$ position of **CCepS-N$^+$(CH$_3$)$_3$–1** with a methoxy group[40,41], the resulting probe (namely **CCepS-N$^+$(CH$_3$)$_3$–3**) can only be hydrolyzed by AmpC and carbapenemase. Finally, **CCS-N$^+$(CH$_3$)$_3$** can only be hydrolyzed by carbapenemase while maintaining silence in the presence of the lower-level $\beta$-lactamases (i.e., BSBL, ESBL, and AmpC). These four probes were incorporated into a BSV sensor to allow rapid and visual identification of BSBL, ESBL, AmpC, and carbapenemase.

This study was set out by preparing the probes. **CCepS-N$^+$(CH$_3$)$_3$–1** was synthesized from a 7-ACA derivative (**1**) whose carboxylic acid was protected by a benzyl ether (Supplementary Fig. 1).

Firstly, 2-(2-aminothiazol-4-yl)acetyl chloride was conjugated to the C7-$\beta$ position of 7-ACA in the presence of bis-trimethylsilylacetamide (BSA) to yield compound **2**. Next, compound **2** underwent nucleophilic substitution of the chlorine with sodium iodide, which was then protected with triphenylphosphine (PPh$_3$) to give compound **3**. Compound **3** was cross-coupled with compound **4** under the catalysis of potassium trimethylsilanolate to produce compound **5**. Finally, the carboxylic acid group in compound **5** was removed using trifluoroacetic acid and anisole to yield **CCepS-N$^+$(CH$_3$)$_3$–1**. **CCepS-N$^+$(CH$_3$)$_3$–2** was synthesized from cefdinir (compound **6**, a classical 3rd-generation cephalosporin antibiotic), which was directly conjugated with compound **7** via the Heck coupling reaction (Supplementary Fig. 2). **CCepS-N$^+$(CH$_3$)$_3$–3** was generated by modifying the C7-$\alpha$ position of cefdinir with a methoxy group, followed by the Heck coupling reaction with compound **7** (Supplementary Fig. 3). These products and related intermediates were characterized by nuclear magnetic resonance (NMR) and mass spectroscopy (MS), as shown in Supplementary Fig. 4–27, Supporting Information.

### Analytical features of CCepS-N$^+$(CH$_3$)$_3$–1, CCepS-N$^+$(CH$_3$)$_3$–2, CCepS-N$^+$(CH$_3$)$_3$–3, and CCS-N$^+$(CH$_3$)$_3$

To evaluate the coverage and specificity of CCepSs for the four subtypes of $\beta$-lactamases, we prepared 11 $\beta$-lactamases, including five carbapenemases (NDM-1, KPC-3, VIM-27, IMP-4, and OXA-48), 2 AmpCs (MOX-1 and CMY-4), 3 ESBLs (SHV-5, TEM-7, and CTX-M-3), and 1 BSBL (TEM-1). These $\beta$-lactamases were synthesized using genetic engineering techniques based on their respective gene sequences. The resulting $\beta$-lactamases were purified using HisTrap Ni columns and identified using SDS-PAGE[34].

We attempted to investigate the color and UV spectral changes caused by the rupture of the lactam ring when the three **CCepS-N$^+$(CH$_3$)$_3$** and **CCS-N$^+$(CH$_3$)$_3$** (10 µM) were incubated with the most broad-spectrum carbapenemase−NDM-1 (3 nM) at 37 °C in phosphate-buffered saline (PBS, pH 7.4). All the solution color of **CCepS-N$^+$(CH$_3$)$_3$** and **CCS-N$^+$(CH$_3$)$_3$** changed immediately from pale yellow to red, with a shift in the maximum absorption bands ($\lambda_{max}$) of **CCepS-N$^+$(CH$_3$)$_3$–1** from 443 nm to 481 nm, **CCepS-N$^+$(CH$_3$)$_3$–2** from 436 nm to 512 nm, **CCepS-N$^+$(CH$_3$)$_3$–3** from 438 nm to 518 nm, and **CCS-N$^+$(CH$_3$)$_3$** from 417 nm to 591 nm (Fig. 2A, i: **CCepS-N$^+$(CH$_3$)$_3$–1**, ii: **CCepS-N$^+$(CH$_3$)$_3$–2**, iii: **CCepS-N$^+$(CH$_3$)$_3$–3**, and iv: **CCS-N$^+$(CH$_3$)$_3$**). The intensity was dependent on the incubation time. To confirm whether the probes were effectively cleaved by NDM-1, liquid chromatograph-mass spectrometer (LC-MS) analysis was performed on the four substrates as well as their corresponding hydrolysis products, as shown in Supplementary Fig. 28-31. The LC retention times of all the hydrolysis products were shorter than their corresponding substrates, and no other LC peaks were observed. These results suggest that the $\beta$-lactam rings in all **CCepS-N$^+$(CH$_3$)$_3$** and **CCS-N$^+$(CH$_3$)$_3$** can be completely cleaved without any intermediate compounds produced during the hydrolysis.

As seen from the spectral changes before and after hydrolysis of the three **CCepS-N$^+$(CH$_3$)$_3$**, the absorbance at 518 nm ($A_{518}$) is greatly improved (~60 folds) within 5 min. Thus, the changes in $A_{518}$ can be indicative of the presence of BSBL, ESBL, and AmpC. In the context of **CCS-N$^+$(CH$_3$)$_3$** before and after hydrolysis, the most exciting finding is that the two bands at 417 and 591 nm were resolved without any spectral overlap. Upon the introduction of NDM-1, more than 120-fold enhancement of the absorbance at 591 nm ($A_{591}$) was achieved within 5 min (Fig. 2A, iv), which can be utilized to sensitively detect carbapenemases.

Subsequently, we evaluated the dynamic interactions of the four visual probes towards BSBLs, ESBLs, AmpCs, and carbapenemases. $A_{518}$ and $A_{591}$ were respectively monitored after incubating the three **CCepS-N$^+$(CH$_3$)$_3$** and **CCS-N$^+$(CH$_3$)$_3$** (10 µM) with the 11 $\beta$-lactamases subtypes (3 nM). In the case of **CCepS-N$^+$(CH$_3$)$_3$–1**, the $A_{518}$ values increased under the treatment of all the 11 $\beta$-lactamases, in which

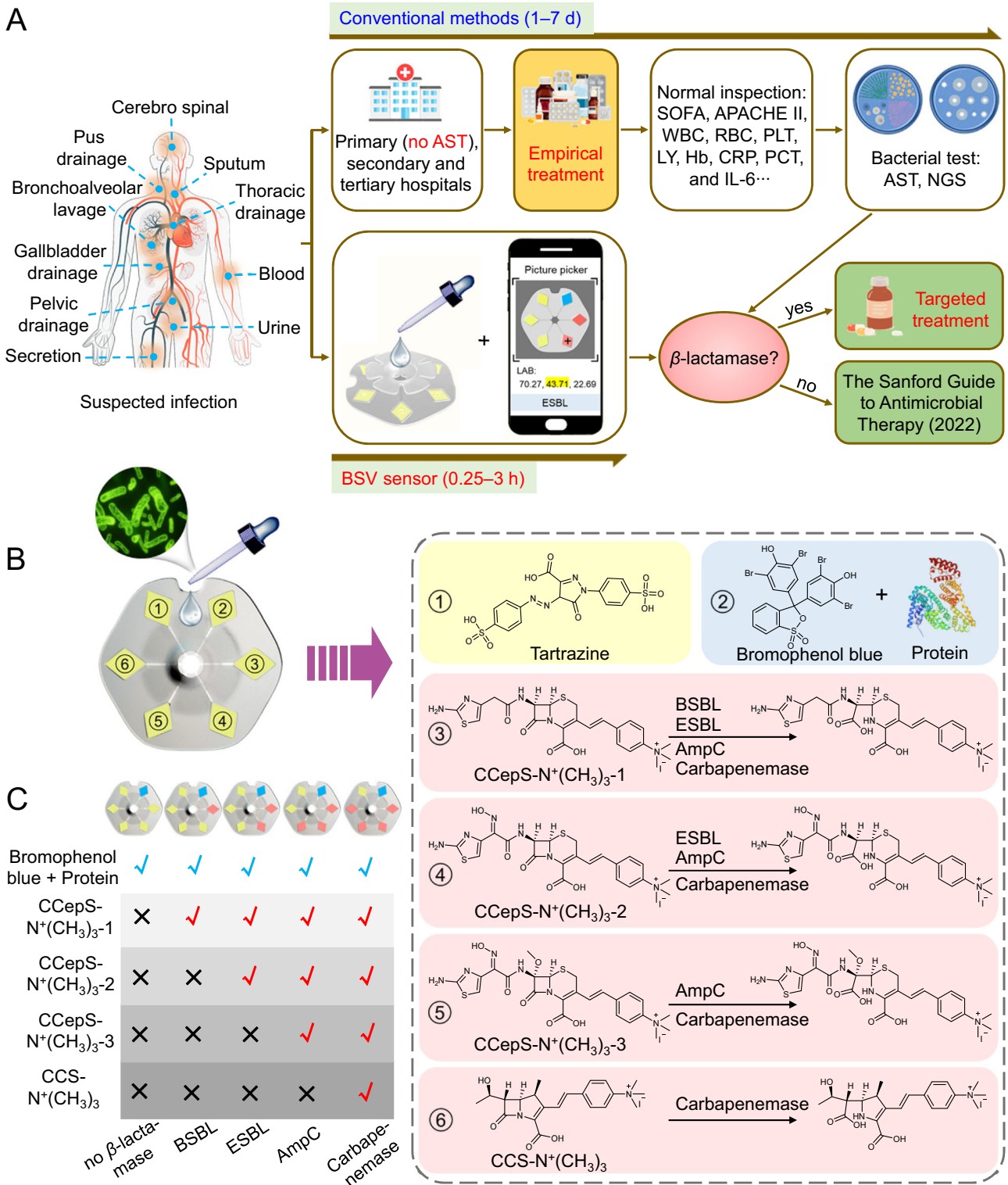

**Fig. 1 | Design of the BSV sensor for rapid detection of antibiotic resistance.**
**A** The management of antibiotic resistance by conventional strategies and the BSV sensor. In most countries, healthcare institutions are commonly categorized into primary (no AST), secondary, and tertiary levels. Typically, suspected infection cases are initially managed to control symptoms in clinical settings based on the physician's experience, and the medication would be adjusted according to the bacterial culturing results several days later. By contrast, the BSV sensor can report the β-lactamase subtypes of various body fluid samples from suspected infection patients within 0.25–3 h. **B** Design of the BSV sensor and working principle of the six independent sample chambers. Chamber 1 is loaded with tartrazine to reflect the quality of the BSV sensor, while chamber 2 makes use of bromophenol blue to indicate the presence of proteins. Chambers 3 to 6 are loaded with the four as-synthesized β-lactamase chromogenic probes to respectively differentiate BSBL, ESBL, AmpC, and carbapenemases. **C** Hydrolysis properties of the four β-lactamase subtypes towards the chromogenic probes.

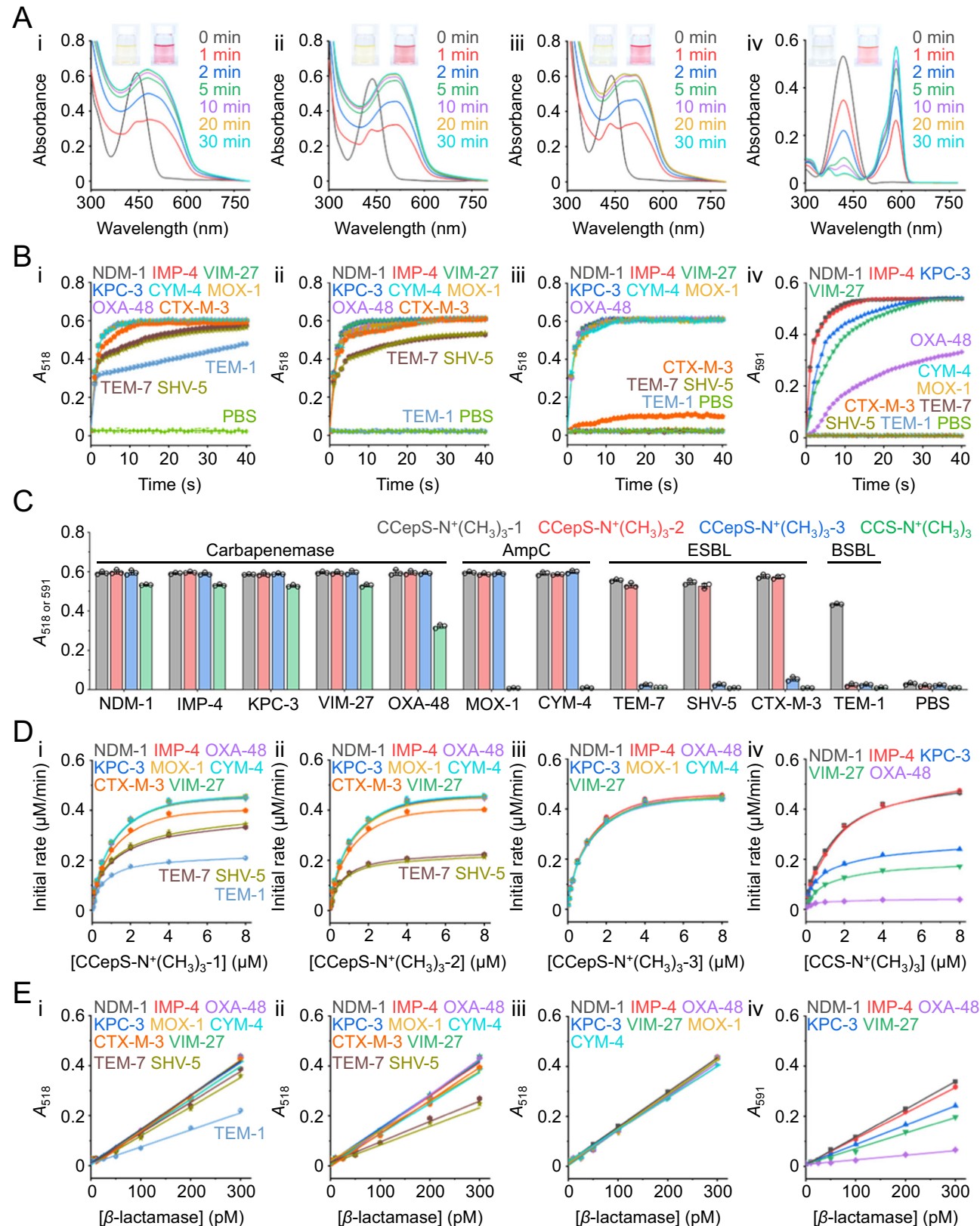

ESBLs (TEM-7, SHV-5, and CTX-M-3) showed a slightly weaker enhancement and BSBL (TEM-1) was the weakest strain but still was able to increase the $A_{518}$ value by 20 folds (Fig. 2B, i). For **CCepS-N$^+$(CH$_3$)$_3$–2**, the $A_{518}$ values were enhanced by ESBLs, AmpCs, and carbapenemases while BSBL was ineffective (Fig. 2B, ii). Neither ESBLs nor BSBLs significantly increased the $A_{518}$ values for **CCepS-N$^+$(CH$_3$)$_3$–3**

(Fig. 2B, iii). Only the five subtypes of carbapenemases increased the $A_{591}$ values for **CCS-N$^+$(CH$_3$)$_3$** (Fig. 2B, iv), whereas the six non-carbapenemases had minimal effect.

To investigate the specificity of **CCepS-N$^+$(CH$_3$)$_3$** and **CCS-N$^+$(CH$_3$)$_3$** towards the 11 $\beta$-lactamase subtypes, the corresponding $A_{518}$ and $A_{591}$ values were collected after incubation for 40 min

**Fig. 2 | Analytical features of CCepS-N⁺(CH₃)₃−1 (i), CCepS-N⁺(CH₃)₃−2 (ii), CCepS-N⁺(CH₃)₃−3 (iii), and CCS-N⁺(CH₃)₃ (iV) towards 11 β-lactamase subtypes.** **A** Spectral and color change of **CCepS-N⁺(CH₃)₃** and **CCS-N⁺(CH₃)₃** (10 μM in PBS, pH 7.4) after incubating with NDM-1 (3 nM) over time. **B** Absorbance change of **CCepS-N⁺(CH₃)₃** (10 μM) at 518 nm ($A_{518}$) and **CCS-N⁺(CH₃)₃** (10 μM) at 591 nm ($A_{591}$) after incubating with 3 nM of carbapenemases (NDM-1, KPC-3, IMP-4, VIM-27, and OXA-48), AmpC (MOX-1 and CYM-4), ESBL (CTX-M-3, SHV-5, and TEM-7), and BSBL (TEM-1) over time. Data represent mean ± s.d., $n = 3$, three technical replicates.

**C** $A_{518}$ of **CCepS-N⁺(CH₃)₃** and $A_{591}$ of **CCS-N⁺(CH₃)₃** at 40 min after incubating with the β-lactamases. Data represent mean ± s.d., $n = 3$, three technical replicates. **D** Michaelis–Menten kinetic plots of the selected β-lactamases towards **CCepS-N⁺(CH₃)₃** and **CCS-N⁺(CH₃)₃**. Data represent mean ± s.d., $n = 3$, three technical replicates. **E** $A_{518}$ of **CCepS-N⁺(CH₃)₃** and $A_{591}$ of **CCS-N⁺(CH₃)₃** (10 μM in PBS, pH 7.4) at 40 min after incubating with various concentrations of the selected β-lactamases. Data represent mean ± s.d., $n = 3$, three technical replicates.

(Fig. 2C). The results showed that **CCepS-N⁺(CH₃)₃−1** exhibited no specificity towards the 11 subtypes of β-lactamases, while **CCepS-N⁺(CH₃)₃−2** demonstrated high specificity towards carbapenemases, AmpC, and ESBL. **CCepS-N⁺(CH₃)₃−3** showed specificity towards carbapenemases and AmpC, while **CCS-N⁺(CH₃)₃** displayed specificity only towards carbapenemases. Based on these results, we believe the combinational analysis of these four probes can enable the visual identification of β-lactamase subtypes.

To better understand the process of β-lactamase hydrolyzing **CCepS-N⁺(CH₃)₃** and **CCS-N⁺(CH₃)₃**, we measured the Michaelis-Menten kinetic parameters, including Michaelis constant ($K_m$) and catalytic constant ($k_{cat}$), by creating a set of Lineweaver-Burk plots. The results showed that β-lactamases could rapidly degrade their corresponding substrate probes (Fig. 2D). Based on the plots, the kinetic efficiencies ($k_{cat}/K_m$) of the enzymes were determined (Supplementary Tables 1–4). The $k_{cat}/K_m$ values of all the **CCepS-N⁺(CH₃)₃** hydrolyzed by their corresponding β-lactamases are comparable to the $k_{cat}/K_m$ values of **CCS-N⁺(CH₃)₃** hydrolyzed by carbapenemases. These visual probes show 1−2 orders of magnitude higher sensitivity than the reported optical probes for β-lactamase detection[20,42,43], confirming that **CCepS-N⁺(CH₃)₃** and **CCS-N⁺(CH₃)₃** possess high reactivity.

Sensitivity is another crucial parameter for probes, especially in real clinical samples. After incubating **CCepS-N⁺(CH₃)₃** and **CCS-N⁺(CH₃)₃** with the β-lactamases at different concentrations (0, 10, 25, 50, 100, 200, and 300 pM) in a 96-well microplate for 40 min, the absorbance at 518 nm (**CCepS-N⁺(CH₃)₃**, $A_{518}$) and 591 nm (**CCS-N⁺(CH₃)₃**, $A_{591}$) was recorded with a microplate reader. By plotting the $A_{518}$ or $A_{591}$ values with the concentrations of β-lactamases, a series of calibration curves were constructed at $3S/k$ (where $S$ is the standard deviation of the blank sample and $k$ is the slope of the standard curve) (Fig. 2E). The limits of detection (LODs) for the β-lactamases were calculated at pM levels with high selectivity (see details in Supplementary Tables 1–4).

Furthermore, the chemical stability of **CCepS-N⁺(CH₃)₃** and **CCS-N⁺(CH₃)₃** were investigated by testing their background hydrolysis via a first-order kinetic curve (Supplementary Fig. 32), the first-order rate constants for the spontaneous hydrolysis reactions of **CCepS-N⁺(CH₃)₃−1**, **CCepS-N⁺(CH₃)₃−2**, **CCepS-N⁺(CH₃)₃−3**, and **CCS-N⁺(CH₃)₃** at room temperature (PBS, pH 7.4) were respectively determined to be $4.8 \times 10^{-6}$, $4.5 \times 10^{-6}$, $4.8 \times 10^{-6}$, and $4.1 \times 10^{-6}$ s⁻¹. Accordingly, their half-lives were estimated to be 40, 43, 40, and 47 h, outperforming the similar β-lactamase probes[25].

### Visual detection of β-lactamase-producing bacterial strains in clinical isolates

After investigating the analytical performance of the as-prepared probes, we wanted to incorporate them into a sensor (Supplementary Fig. 33) for rapid discrimination of β-lactamase subtypes. To standardize the testing process, various concentrations of the probes (5 μL) were dropped onto filter papers, which were then installed on a polyethylene sensor (Fig. 3A). Tartrazine (a yellow dye) and bromophenol blue (a blue dye) were respectively set as negative and positive controls. By comparing the difference in color changes before and after incubating with NDM-1, the optimal concentrations of tartrazine,

bromophenol blue, **CCepS-N⁺(CH₃)₃−1**, **CCepS-N⁺(CH₃)₃−2**, **CCepS-N⁺(CH₃)₃−3**, and **CCS-N⁺(CH₃)₃** used for fabricating the BSV sensors were determined to be 0.1, 1.0, 1.8, 1.8, 1.8, and 2.0 mM, respectively (Supplementary Fig. 34).

Additionally, we have developed a foldable shadowless lamp that was coupled with a smartphone to obtain stable images regardless of environmental factors (Supplementary Fig. 35). A color recognizer software was installed in the smartphone to quantitatively analyze the color changes of the sensors. The color recognizer offers 8 color modes: digital equipment corporation (DEC), hexadecimal (HEX), alpha-red-green-blue (ARGB), cyan-magenta-yellow-key (CMYK), CIE-LAB (Lab), hue-saturation-value (HSV), hue-saturation-lightness (HSL), and luma-chroma (YUV)[44,45]. After investigating the negative, weak positive, and positive results on three randomly-selected sensors (Supplementary Fig. 36), we found that CIELAB (Lab) is the most suitable mode to differentiate the results by converting the colors into a* and b* values. This is because in the Lab mode, L* represents the lightness from black to white, a* from green to red, and b* from blue to yellow. The a* and b* values could report the yellow-to-red and yellow-to-blue changes, respectively, without the need for professional graphics software such as Photoshop.

After making the necessary preparations, we investigated the selectivity and sensitivity of the BSV sensor for the common clinical bacterial isolates, including *Escherichia coli* (*E. coli*), *Klebsiella pneumoniae* (*K. pneumoniae*), and *Morganella morganii* (*M. morganii*). The clinical isolates were promptly frozen and sonicated to release β-lactamases from the bacteria. Firstly, PBS (blank control), Luria-Bertani (LB) medium, and β-lactamase-negative *E. coli* lysate were separately pipetted onto the sampling hole of the BSV sensor. When added PBS, the color of all the chambers did not change (Fig. 3B and Supplementary Fig. 37). In contrast, LB medium and β-lactamase-negative *E. coli* lysate turned the papers blue only in chamber 2, which was attributed to the interactions between the proteins and the loaded bromophenol blue. Note that the color in chambers 3-6 did not change, indicating that no false-positive results were generated. Subsequently, the lysates of *E. coli* (TEM-1), *K. pneumoniae* (SHV-18), *M. morganii* (DHA-1), and *E. coli* (NDM-1) were sequentially tested to validate the performance of the BSV sensor. The lysate of *E. coli* (TEM-1) only changed the color of chamber 3, while the lysate of *K. pneumoniae* (SHV-18) caused chambers 3 and 4 to turn red. The lysate of *M. morganii* (DHA-1) caused chambers 3-5 to turn red, and the lysate of *E. coli* (NDM-1) caused all detection chambers 3-6 to change color, fully agreeing with the expected results. The BSV sensor showed excellent selectivity in discriminating the β-lactamase subtypes.

We then tested the lowest detectable concentrations for the isolates with diverse β-lactamase subtypes. Compared to other β-lactamases, the enzymatic activity of TEM-1 is lower[31]. Therefore, the lowest detectable concentration for *E. coli* (TEM-1) was determined to be $10^5$ CFU/mL (100 CFU), while the others were determined to be $10^4$ CFU/mL (10 CFU) (Fig. 3B). Using the Lab color model to quantify the sensor results, the values fully reflect the resistance types and levels of β-lactamases. As the β-lactamase levels increase, the a* values show a close correlation with the positive results. Meanwhile, the b* values reflect the positive control in chamber 2 (Fig. 3C).

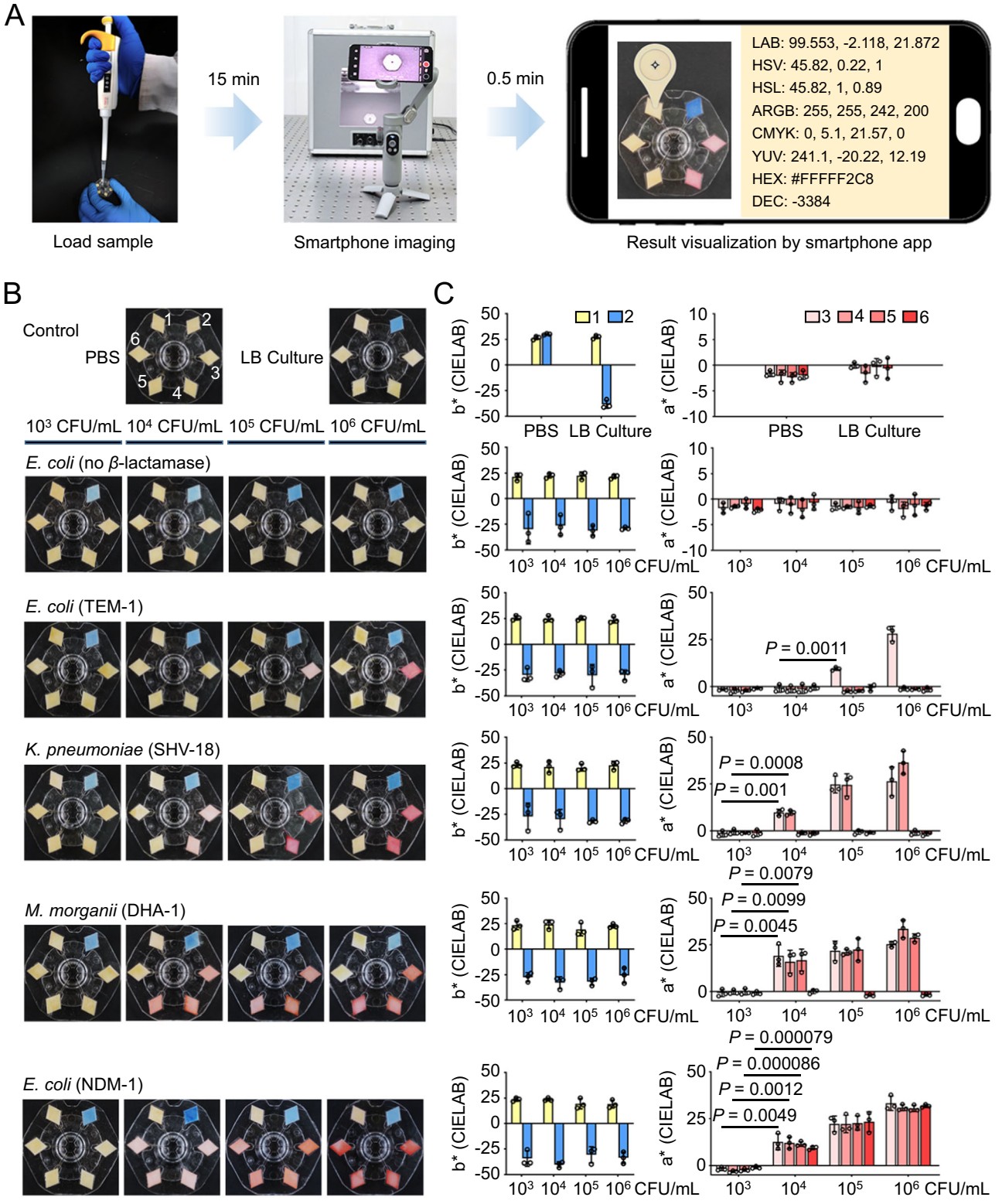

**Fig. 3 | Using the BSV sensor to detect the β-lactamase levels in clinically isolated bacteria. A** Flowchart of the BSV sensor testing process. The bacteria samples were dropped into the sampling hole of the sensor, followed by capturing the image and analyzing the color information with a smartphone. **B** The BSV sensors were used for detecting the clinical isolates of five different bacterial lysates at various concentrations from $10^3$ to $10^6$ CFU/mL. The BSV sensor images were captured after incubation at 15 min. **C** The color changes of the BSV sensor depicted

in (**B**) were quantified using a smartphone equipped with color recognition software. The $b^*$ value and $a^*$ value, representing the transitions from yellow to blue and from green to red respectively, are recorded in the CIELAB (Lab) color model, which is a standardized color space used for colorimetry and spectroscopy analysis in biomedical research (paired two-tailed Student's $t$ test). Data represent mean ± s.d., $n = 3$, three technical replicates.

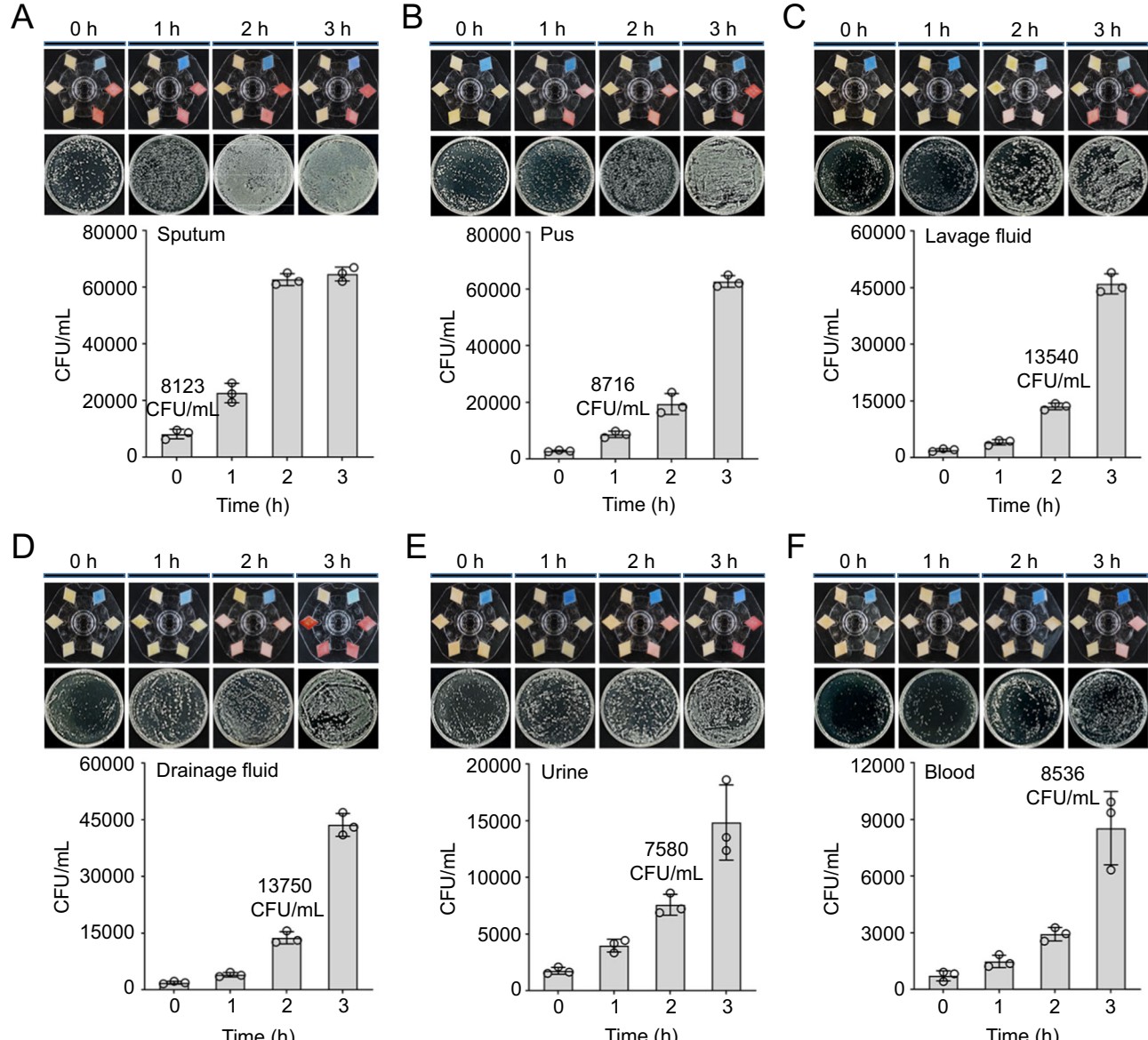

**Fig. 4 | Using the BSV sensor to detect bacterial isolates in various body fluid samples. A** The sputum sample count on a culture medium rich in cefotaxime/clavulanic acid was 8123 CFU/mL, which can be directly detected without cultivation ($n = 3$ biologically independent samples). Error bars: mean ± s.d. **B** The pus sample count on a culture medium rich in cefotaxime/clavulanic acid was 2853 CFU/mL, which required cultivation for 1 h to proliferate to 8716 CFU/mL before testing ($n = 3$ biologically independent samples). Error bars: mean ± s.d. **C** The lavage fluid sample count on a culture medium rich in cefoxitin was 2020 CFU/mL, which requires cultivation for 2 h to proliferate to 13,540 CFU/mL before testing ($n = 3$ biologically independent samples). Error bars: mean ± s.d. **D** The drainage fluid sample count on a culture medium rich in meropenem was 1863 CFU/mL, which requires cultivation for 2 h to proliferate to 13750 CFU/mL before testing ($n = 3$ biologically independent samples). Error bars: mean ± s.d. **E** The urine sample count on a culture medium rich in cefotaxime/clavulanic acid was 1756 CFU/mL, which required cultivation for 2 h to proliferate to 7580 CFU/mL before testing ($n = 3$ biologically independent samples). Error bars: mean ± s.d. **F** The blood sample count on a culture medium rich in cefotaxime/clavulanic acid was 723 CFU/mL, which required cultivation for 3 h to proliferate to 8536 CFU/mL before testing ($n = 3$ biologically independent samples). Error bars: mean ± s.d.

## Detection of β-lactamases in the body fluids of patients with infections at different sites

Sepsis can be caused by infections in any part of the body[46]. Among hospitalized patients, the most common infections leading to sepsis are lower respiratory tract infections, followed by abdominal infections, bloodstream infections, vascular infections, and urinary tract infections[8]. In this study, we employed this BSV sensor to detect the β-lactamase-expressed bacteria in commonly encountered samples such as blood, urine, tissue irrigation fluid, and wound drainage fluid in clinical practice.

The abundance of bacterial isolates varies in the above-mentioned biofluids. For example, the concentrations of antibiotic-resistant

bacterial isolates in unprocessed clinical sputum samples are typically above $10^4$ CFU/mL levels, which are higher than the detection limits of the BSV sensor. Therefore, the isolates in the sputum of the patients with pulmonary infections can be directly detected, without resorting to cultivation (Fig. 4A). However, for other biofluids, such as wound secretions, lavage fluid, drainage fluid, blood, and urine, the antibiotic-resistant bacterial counts are generally below $10^4$ CFU/mL, which requires various periods of cultivation to increase the bacteria concentrations to fulfill the detection sensitivity of the visual probes (Fig. 4 and Supplementary Fig. 38). For example, the antibiotic-resistant bacterial isolates (2853 CFU/mL) in a pus sample required 1 h of cultivation to increase the counts to 8716 CFU/mL for testing. For lavage

fluid, drainage fluid, and urine samples, they needed 2 h of cultivation to increase their counts from 2020, 1863, and 1756 CFU/mL to 13540, 13750, and 7580 CFU/mL, respectively. With respect to the blood sample, its antibiotic-resistant bacterial count was 723 CFU/mL, which required a longer time (3 h) of cultivation to increase its count to 8536 CFU/mL for visual detection. Note that various antimicrobial drugs were added to the culture media to kill the non-resistant bacteria, retaining the resistant ones for the following detection. Although a few hours of cultivation were required for some clinical samples, the BSV sensor matches the clinical sensitivity and specificity offered by AST that need days to complete.

### Visual identification of β-lactamase subtypes in the body fluids of patients with bacterial infections

To assess the clinical potential of the BSV sensor for the rapid identification of β-lactamase subtypes, we collected 100 body fluids from the patients in the critical care department at the First Central Hospital of Tianjin (a comprehensive tertiary grade-a hospital). The patients had suffered various infectious diseases, including pulmonary, abdominal, organ, and traumatic infections. Based on the physicians' empirical prescription before obtaining the culture-based results, only 43 patients received appropriate treatment, 41 patients received ineffective treatment and 16 patients suffered from severe drug abuse (Supplementary Data 1). It is, therefore, vital to classify the subtypes of antibiotic resistance, which could enable personalized drug use.

With the BSV sensor, 4 cases of BSBL, 20 cases of ESBL, 10 cases of AmpC, and 34 cases of carbapenemase resistance were detected, based on which the β-lactamase resistance rate was determined to be 68%. We verified our detection results with RT-PCR, which provides the genetic information of the overexpressed β-lactamases. The RT-PCR results agreed well with our BSV sensor results (Fig. 5A and Supplementary Figs. 39–41). It is worth mentioning that RT-PCR reported false-positive ESBL results for patients NO.18, NO.36, NO.42, and NO.56 (as validated by the gold-standard AST results in Supplementary Fig. 42), which is most likely due to the fact that some BSBL and ESBL genes show similar sequences that are difficult to be discriminated by RT-PCR[38,47]. These results further demonstrate the high effectiveness of the BSV sensor for accurate identification of β-lactamases.

The 100 patients involve various infectious diseases, including 68 cases of pulmonary infections, 14 cases of urinary tract infections, 7 cases of abdominal infections, 4 cases of liver abscesses, 2 cases of gallbladder abscesses, 1 case of neck abscess, 1 case of sinusitis, 1 case of pancreatitis, 1 case of pelvic inflammatory disease, and 1 case of leg inflammation. These clinical specimens include 56 sputum samples, 14 urine samples, 9 bronchoalveolar lavage fluids, 7 peritoneal drainage fluids, 4 hepatic drainage fluids, 2 gallbladder drainage fluids, 2 blood samples, 1 pleural effusion, 1 neck abscess fluid, 1 pelvic effusion, 1 sinus lavage fluid, 1 pancreatic drainage fluid, and 1 leg abscess fluid (Fig. 5B). The BSV sensor provided results in 0.25–3 h dependent on the sample types, whereas AST (the traditional testing method used in hospitals) typically takes at least 48 h and in some cases up to 144 h (Fig. 5C). Obviously, the BSV sensor can be utilized to monitor the bacterial infection process, allowing for timely adjustment of medication plans. In contrast, the AST results can only reflect the infection status of the sample at the time of collection, which can not report the development of bacterial infections in real time. These findings highlight the benefits of the BSV sensor in aiding timely and accurate treatment against bacterial infections.

The detection accuracy of the sensor results was validated by the Kirby-Bauer paper dispersion method (K-B method) for AST and RT-PCR verification recommended by the 2021 version of the CLSI Performance Standards for Antimicrobial Susceptibility Testing[48]. The results of the K-B method show that cephalothin tablets, cephalothin/clavulanic acid tablets, cefoxitin tablets, and meropenem tablets, which have been treated with the patient body fluids, exhibit varying

degrees of inhibitory effects on *E. coli*. According to the 2021 edition of the Clinical and Laboratory Standards Institute (CLSI) guidelines, an inhibition zone diameter of the cephalothin disk less than 22 mm is indicative of a positive result for BSBL, while that of the cefotaxime/clavulanic acid disk larger than the cephalothin group for 5 mm indicates a positive result for ESBL. Further, the positive results for AmpC and carbapenemase in the samples are respectively indicated by cefoxitin and meropenem disk inhibition zone diameters of less than 18 mm and 19 mm[48]. A total of 4 samples were found to be positive for BSBL, 20 for ESBL, 10 for AmpC, 34 for carbapenemase, and 32 samples were negative for β-lactamase, consistent fully with the results obtained by the BSV sensor (Supplementary Fig. 42 and Supplementary Table 5). Additionally, 96% of the RT-PCR results were in accordance with the BSV sensor results. Due to the fact that the primer sequences for TEM-1, TEM-2, and SHV-1 are included in both TEM and SHV, the 4 BSBL-positive samples (labeled patients NO.18, NO.36, NO.42, and NO.56 with red) might also be mistakenly detected as ESBL-positive (Fig. 5D).

### Using BSV sensor to enhance the accuracy of antibiotic therapy

In the clinic, due to the lack of rapid POCT assays to identify bacterial resistance, physicians frequently administer the initial treatment against bacterial infections based on their medical expertise, known as empirical medication. In general, the accuracy of empirical medication is less than 50%[10]. Physicians have to adjust their prescriptions after obtaining the AST results that call for days of cultivation. Since β-lactamase is the major contributor to bacterial resistance and the therapeutic regimens are closely associated with the β-lactamase subtypes (Fig. 6A), the rapid detection of β-lactamase and accurate discrimination of their subtypes is essential for precision antibiotic therapy.

To validate the utility of the BSV sensor for precision clinical therapy, the empirical therapeutic regimens for the above 100 infection patients were analyzed and compared to those assisted by the BSV sensor. The empirical medication was shown in Supplementary Data 1, column 6; after acquiring the AST results, the therapeutic regimens were adjusted and moved to the so-called targeted medication (Supplementary Data 1, column 7). Obviously, a large number of under-treatment and over-treatment cases are found in empirical medications. A typical under-treatment case is patient NO.1. This patient did not show the hallmarks of bacterial infections and thus had not received treatment. However, after 48-hour cultivation, the AST results indicated that the patient had been infected by bacteria and Tigecycline was the recommended antibiotic. In parallel, our BSV sensor results show that patient NO.1 was carbapenem-resistant (Fig. 5A), for which Tigecycline is one of the effective antibiotics in the toolbox (Fig. 6A). A typical case of over-treatment is patient NO.7, who had been empirically treated with Polymyxin B and Levofloxacin. However, this patient was β-lactamase-negative, as verified by AST and our BSV sensor results, meaning that patient NO.7 was over-treated by empirical medication. We employed the BSV sensor to optimize the empirical drug consumption for the 100 cases (Fig. 6B). It was encouraging that the accuracy of antibiotic therapy can be increased from 43% (empirical treatment) to 83% (BSV-assisted treatment) (Fig. 6C).

We found that some patients were infected by bacteria and fungi simultaneously. For instance, patient NO.28 was empirically diagnosed as fungi-infectious and was only prescribed with an antifungal medicine—Caspofungin. However, our BSV sensor reported that the patient was also infected by the AmpC-resistant bacteria, which was validated by AST. The administration of antifungal or antibacterial drugs alone was unable to cure the patient. It is worth noting that in a few cases, the patients were infected only by fungi. Since our BSV sensor can only report bacterial infections, it is insufficient for helping physicians make correct prescriptions. Fortunately, in clinical practice, fungal infection

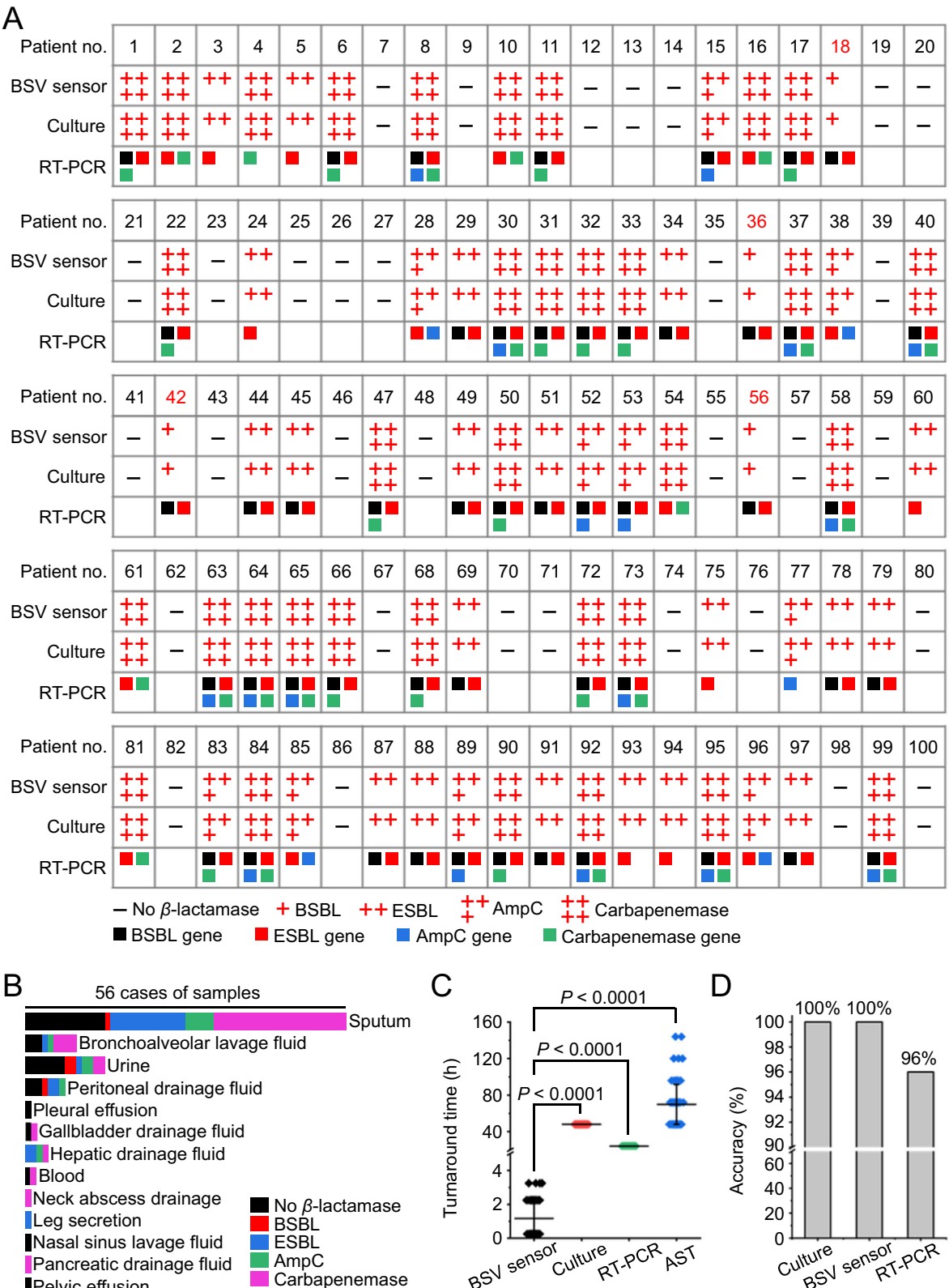

**Fig. 5 | The BSV sensors were used to identify the β-lactamase subtypes in 100 clinical samples. A** The BSV testing results were verified by bacterial culture and RT-PCR. **B** The clinical samples involve 13 kinds of body fluids, among which the sputum samples from pneumonia patients rank the top one, reaching 56 cases. Various β-lactamase subtypes were found in these samples. **C** Average ± s.d. turnaround time of the BSV sensor, bacterial culture, and RT-PCR, which were compared with the AST time in the clinical data (Supplementary Data 1) (paired two-

tailed Student's *t* test, $P < 0.0001$, $n = 100$ biologically independent samples). Error bars: mean ± s.d. **D** Accuracy of β-lactamase diagnosis obtained from bacterial culture, BSV sensor, and PT-PCR. The results obtained from the BSV sensor were found to have the same clinical accuracy to the traditional culture-based β-lactamase diagnostic technique and were slightly better than those obtained from RT-PCR.

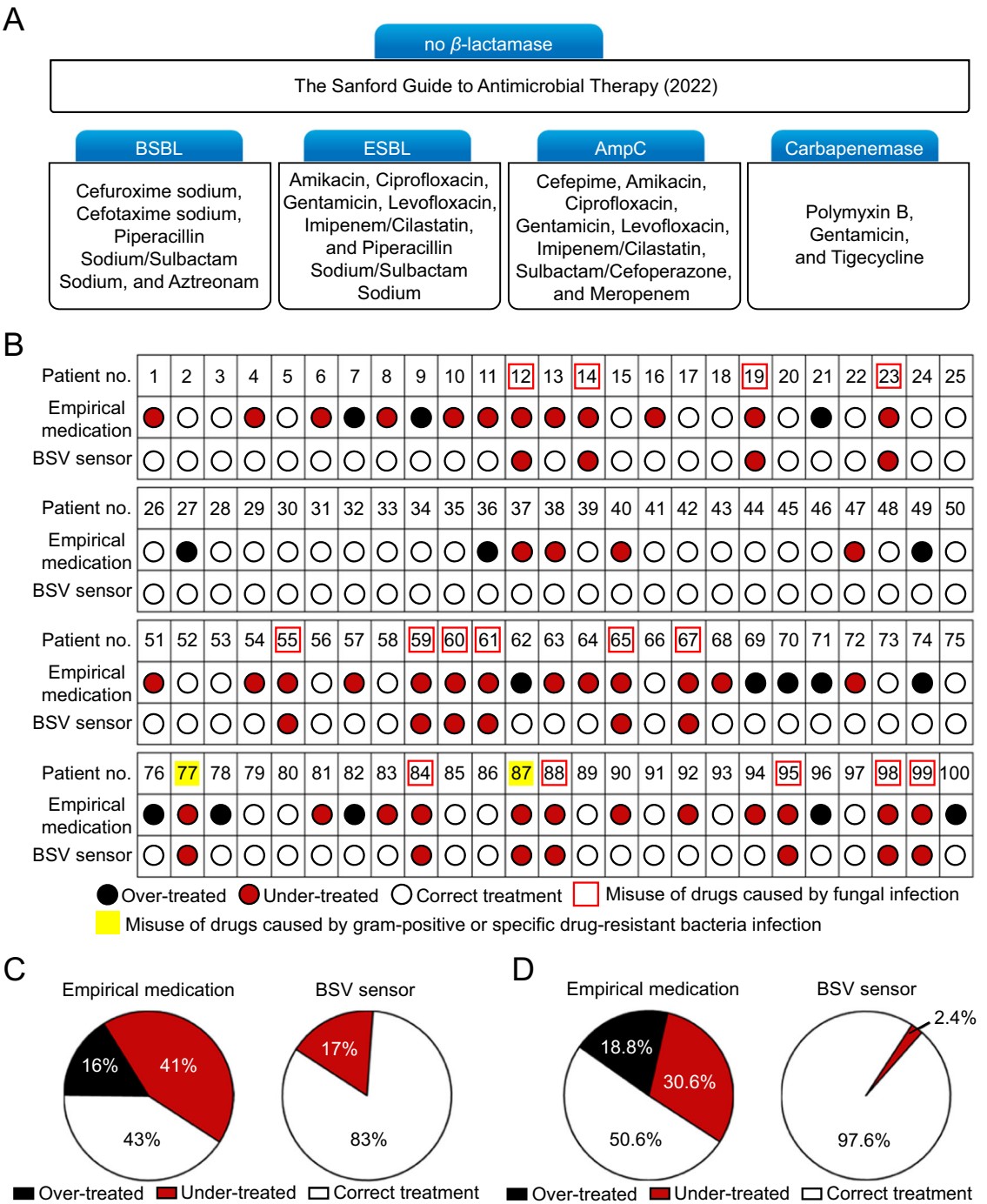

**Fig. 6 | The BSV sensor-assisted precision antibiotic therapy that was compared with empirical medication. A** A list of medications classified by $\beta$-lactamase subtypes. Note that the cases in the absence of $\beta$-lactamases can be referred to the Sanford Guide to Antimicrobial Therapy (2022). **B** The prescriptions that were made empirically and the same ones were made according to the BSV sensor results. **C** The BSV sensor can eliminate antibiotic overuse, reduce the proportion of under-treated cases from 41% to 17%, and enhance the correct treatment from 43% to 83%. **D** By excluding fungal interference, the BSV sensor could further reduce the fraction of under-treated cases from 30.6% to 2.4% while increasing the correct treatment from 50.6% to 97.6%.

cases can be readily found by the physicians' initial assessment, and their treatment can be referred to the Sanford Guide to Antimicrobial Therapy. Therefore, we excluded the fungal infections from the 100 testing cases (indicated by the red boxes in Fig. 6B), after which the accuracy of antibiotic therapy was enhanced from 50.6% empirically to an astounding 97.6%, yet there were still 2 cases misdiagnosed (patients NO.77 and NO.87 labeled with yellow background). A careful examination revealed that patient NO.77 was infected by *Staphylococcus aureus*, a Gram-positive bacterium that Sulbactam/

Cefoperazone was ineffective. The infectious bacteria in patient NO. 87 was *Stenotrophomonas maltophilia*, which was naturally resistant to carbapenem antibiotics, making the ESBL-positive result offered by the BSV sensor helpless to physicians. For patient NO. 77, a Gram stain test can be used to identify the Gram-positive/negative bacteria, which could be combined with the sensor results to help physicians make effective prescription. In clinical practice, the cases infected by Gram-positive bacteria that simultaneously overexpress AmpC enzymes should be prescribed with vancomycin[49]. The expression of

*Stenotrophomonas maltophilia* in the infectious bacteria of patient NO. 87 can be identified by a qPCR kit. Since even carbapenem antibiotics are ineffective to this infection, patient NO. 87 could be given non-$\beta$-lactamase antibiotics such as levofloxacin, minocycline, or cotrimoxazole[50].

## Discussion

In this study, we devised and synthesized a series of small-molecule chromogenic probes with high sensitivity and specificity against the corresponding $\beta$-lactamase subtypes, including BSBL, ESBL, AmpC, and carbapenemases. The probes were loaded onto filter paper sheets and then incorporated into a BSV sensor, enabling visual identification and semi-quantitative detection of $\beta$-lactamase variants. A wide range of body fluid samples, including sputum, blood, urine, organ drainage fluid, and tissue lavage fluid, were used to test the efficacy of this sensor, which was confirmed by AST and RT-PCR. As a developed POC diagnostic device that can quickly screen and categorize $\beta$-lactamase-producing bacteria, this sensor overcomes the bottlenecks of slow bacterial culture testing, cumbersome operation, and inability to monitor bacterial resistance in real-time, potentially revolutionizing the way of antibiotic treatment.

The BSV sensor we developed is specifically made for patients who are confined to an intensive care unit (ICU) or who are facing life-threatening medical emergencies brought on by bacterial infections. At present, primary antibiotic therapy mainly depends on the physicians' empirical assessment of the infection situation and medication knowledge, by which the accuracy of antibiotic therapy is overall less than 50%[10]. With this sensor, the $\beta$-lactamase-producing bacteria in real clinical samples were identified within 0.25−3 h, which dramatically enhanced the accuracy of antibiotic therapy from 48% to 83%, and further from 50.6% to 97.6% after eliminating fungal interference. The BSV sensor thus provides possibilities for precision antibiotic therapy, especially in resource-poor regions where expensive facilities and skilled personnel are lacking for microbial culture. In addition to assisting antibiotic therapy, this sensor may help to reduce the overuse and dissemination of antibiotics in both hospital and community settings. In the future, we believe the BSV sensor can be extended to home testing for directing antibiotic uses.

While the BSV sensor is effective in the diagnosis of bacterial infections, they are unable to detect fungal infections. Fungi also pose a significant threat to the health of infected patients, and the initial medication for urgent fungal infections relies on the physician's experience. Urgent fungal testing can only be performed using RT-PCR or test kits that correlate with the source of the infections; otherwise, the physician must wait for the results of the microbiological identification from the laboratory before administering treatment. Pulmonary fungal infections, specifically those caused by *Aspergillus niger*, can be identified through the presence of D-glucosamine[51]. The diagnosis of invasive fungal infections (IFIs) relies on detecting serum markers such as 1,3-β-D-glucan (BDG), mannan (Man), and galactomannan[52]. To enable the rapid detection of fungal infections, we need to develop sensitive molecular probes that can specifically respond to these markers. Moving forward, we will develop fungal probes to couple with this BSV sensor for comprehensive analysis of microbial infections, fostering practical applications in guiding timely antimicrobial therapeutics.

## Methods
### Ethical statement
The study was approved by Nankai University Institutional Review Board (NKU IRB). All human samples were acquired and handled according to the protocols approved by the Scientific Ethical Committee of the First Central Hospital of Tianjin (Approval No. 2023DZX12). All patients' age, sex, disease details and medications are published with the patients' written consent. Informed consent was obtained, and no compensation was provided for all research participants.

### General information
All chemical reagents were purchased from commercial suppliers and used without further purification. The chemical reactions were carried out in dry glassware under an argon atmosphere. Analytical reagent (AR)-grade solvents were used throughout all the reactions. All the dry and absolute solvents were prepared according to standard laboratory procedures and were stored over proper drying agents under an argon atmosphere. Analytical thin-layer chromatography (TLC) was performed using Yantai Huanghai HSGF 254 silica gel plate (0.2 mm silica layer with fluorescent indicator). All the glass instruments used in organic synthesis experiments were purchased from Synthware. Luria-Bertani (LB) culture media and dishes were purchased from Sangon Biotech and sterilized by high-pressure steam prior to use. Some strains were obtained from Prof. Kui Zu at China Agricultural University, including *Escherichia coli* (NDM-1), and *Escherichia coli* (ATCC 25922); the others were purchased from commercial sources (Nanjing Lezhen Biotechnology Co Ltd), including *Escherichia coli* (TEM-1, ATCC 35218), *Klebsiella pneumoniae* (SHV-18, ATCC 700603) and *Morganella morganii* (DHA-1, ATCC 25830). Bacteria culture was completed in a Thermo Scientific Heratherm constant temperature incubator. UV-vis absorption spectra were recorded by a Shimadzu UV-vis spectrophotometer. LC-MS determination was performed on a Shimadzu LC-MS/MS system equipped with a LC-20AT gradient pump and an online diode UV-vis detector. The structures of the compounds were determined by nuclear magnetic resonance (NMR) and mass spectrometry (MS). The NMR shift ($\delta$) is given in units of $10^{-6}$ (ppm). NMR was measured on a Bruker Ascend™ 400 nuclear magnetic instrument in deuterated dimethyl sulfoxide (DMSO-$d_6$), deuterated chloroform (CDCl$_3$), and deuterated methanol (CD$_3$OD). Tetramethylsilane (TMS) was used as the internal standard. The following abbreviations are used for the multiplicity of the NMR signal: s (singlet), brs (broad), d (doublet), t (triplet), m (multiplet). The coupling constants are listed in J values and measured in Hz.

### Fabrication of the BSV sensor
The production process of the sensor involves four essential steps: structural design, micro/nano fabrication, production of test paper, and sensor assembly. The initial step entails designing the sensor in a hexagonal shape using L-Edit software. The sensor design incorporates six independent diamond-shaped chambers, which are connected to the inlet through dedicated channels. The chambers have ventilation holes that ensure smooth sample delivery. Specific dimensions for the chambers are illustrated in Supplementary Fig. 33. For micro/nano fabrication, transparent polyethylene plastic is preferred due to its excellent properties. Next, tartrazine (0.1 mM, 5 μL), bromophenol blue (1 mM, 5 μL), **CCepS-N⁺(CH₃)₃−1** (1.8 mM, 5 μL), **CCepS-N⁺(CH₃)₃−2** (1.8 mM, 5 μL), **CCepS-N⁺(CH₃)₃−3** (1.8 mM, 5 μL), and **CCS-N⁺(CH₃)₃** (2 mM, 5 μL) were dissolved in chloroform, which were respectively dropped onto the diamond-shaped filter paper sheets. Finally, the paper sheets were assembled onto the sensor, completing the process of BSV sensor production.

### LC-MS analysis of CCepS-N⁺(CH₃)₃−1, CCepS-N⁺(CH₃)₃−2, CCepS-N⁺(CH₃)₃−3, CCS-N⁺(CH₃)₃ and their hydrolyzed products
The molecular weights of the compounds were determined using a Shimadzu LC-MS system equipped with an LC-20AT gradient pump and an online diode UV-vis detector. A ShimNex WR C18 column (4.6 i.d. ×250 mm length, 5 μm particle size) was used for chromatography that was operated at 25 °C. The samples were prepared at a concentration of 1 μM, with an injection volume of 50 μL and a flow rate of 1.2 mL/min. Detection was carried out at a wavelength of 254 nm. The

mobile phase consisted of MeCN and 0.1% trifluoroacetic acid solution. The entire procedure was completed within a running time of 25 min.

## Determination of the detection limits of CCepS-N⁺(CH₃)₃−1, CCepS-N⁺(CH₃)₃−2, CCepS-N⁺(CH₃)₃−3 and CCS-N⁺(CH₃)₃

Aliquots of the four different probes (**CCepS-N⁺(CH₃)₃−1, CCepS-N⁺(CH₃)₃−2, CCepS-N⁺(CH₃)₃−3, and CCS-N⁺(CH₃)₃**) were mixed with varying concentrations (0, 10, 25, 50, 100, 200, and 300 pM) of $\beta$-lactamases in pH 7.4 PBS buffer at 37 °C for 30 min. The incubation was performed in a sterile 96-well microplate, and the absorbance values at 518 nm ($A_{518}$) and 591 nm ($A_{591}$) were measured using a microplate reader during the incubation. Specific calibration curves were constructed by plotting $A_{518}$ and $A_{591}$ values against the concentrations of $\beta$-lactamases, by which the slopes ($k$) were obtained. The detection limits of $\beta$-lactamases were determined using 3-fold standard deviations ($S$) of six parallel blank samples divided by the slope ($k$). This allowed for the accurate determination of the minimum concentration of $\beta$-lactamases that could be detected in the samples.

## $\beta$-lactamases kinetics

We determined the kinetics data of $\beta$-lactamases by the Michaelis-Menten model. Aliquots of the 11 $\beta$-lactamases (0.2 nM) were incubated at 37 °C with 1 mL the serially diluted **CCepS-N⁺(CH₃)₃−1, CCepS-N⁺(CH₃)₃−2, CCepS-N⁺(CH₃)₃−3 and CCS-N⁺(CH₃)₃**, whose final concentrations were set to be 0, 0.0625, 0.125, 0.25, 0.5, 1, 2, 4, and 8 μM in pH 7.4 PBS. The $A_{518}$ of (**CCepS-N⁺(CH₃)₃−1, CCepS-N⁺(CH₃)₃−2**, and **CCepS-N⁺(CH₃)₃−3**), and $A_{591}$ of (**CCS-N⁺(CH₃)₃**) were determined every 30 s, and the initial rates ($v$) were determined by the Michaelis-Menten equation.

$$v = \frac{V_{max}[S]}{K_m + [S]} \tag{1}$$

Where $V_{max}$ is the maximal reaction rate at which the $\beta$-lactamase is sufficient to hydrolyze the substrate; [S] values are the substrate concentration. Based on the Michaelis-Menten equation, kinetic curves were obtained. To accurately determine the $K_m$ values, Lineweaver-Burk diagrams were drawn by using the double reciprocal method, where their $K_m$ value was determined according to the y-axis intercept. Finally, the $k_{cat}$ and $k_{cat}/K_m$ values were calculated based on the $V_{max}$ and enzyme concentrations ($k_{cat} = V_{max}/0.2$). To guarantee the reliability of the outcomes, three separate replications were carried out.

## Visual detection of the clinical isolates

We selected five clinical isolates, including *E. coli* (no $\beta$-lactamase), *E. coli* (TEM-1), *K. pneumoniae* (SHV-18), *M. morganii* (DHA-1), and *E. coli* (NDM-1), to evaluate the performance of the sensor. The strains were cultured in 20 mL of LB medium at 37 °C until the optical density (OD₆₀₀) reached 0.8–0.9. Then, 100 μL of 0.5% Triton X-100 was added to 1 mL of bacterial solution and incubated for 10 min. The mixture was quickly frozen in liquid nitrogen and sonicated for 2 min. 1 μL of each sonicated bacteria lysate was diluted to be 20 μL, which was then pipetted onto the sensor and incubated at 37 °C for 5 min. To further investigate the sensitivity of the sensor, the sonicated bacterial lysates were serially diluted in a 96-well plate to final concentrations of $1 \times 10^3$, $1 \times 10^4$, $1 \times 10^5$, and $1 \times 10^6$ CFU/mL.

## Rapid POC detection of diverse body fluid samples

Diverse body fluid samples were collected from 100 infected patients, including 56 sputum specimens, 14 urine specimens, 9 bronchoalveolar lavage fluids, 7 peritoneal drainage fluids, 4 hepatic drainage fluids, 2 gallbladder drainage fluids, 2 blood specimens, 1 pleural effusion specimen, 1 neck abscess fluid, 1 pelvic effusion specimen, 1 nasal sinus lavage fluid, 1 pancreatic drainage fluid, and 1 leg abscess fluid. Except for sputum samples that could be directly tested, the other samples were cultured for 1–3 h to increase the bacterial concentrations above the detection sensitivity of the sensor before testing. Then, 100 μL of 0.5% Triton X-100 was mixed with 1 mL of the patient sample and incubated for 10 min. The mixture was quickly frozen in liquid nitrogen and sonicated for 2 min. Then, 20 μL of each sonicated bacterial lysate was spotted onto the sensor and incubated at 37 °C for 5 min. The photos of the sensor were taken by a smartphone under a shadowless lamp. The color recognition software was used to process the images by converting the color into the Lab color model, resulting in a quantifiable a* value. We chose the a* channel because green is the complementary color of red and red is the endpoint color of the reaction (green to red, −128 to 127).

## Kirby-Bauer paper dispersion method

The 100 diversified body fluid samples were cultured on LB media overnight to produce pure colonies. The colonies were then picked up with an inoculated loop and transferred into 2 mL of Tryptic Soy Broth (TSB) for overnight culture. Subsequently, 1 μL of the cultured colonies were added to another 2 mL of TSB and mixed for 10–15 s to produce bacterial suspensions. Sterile paper disks containing 30 μg cephalothin, 40 μg cephalothin/clavulanic acid, 30 μg cefoxitin, and 10 μg meropenem were dipped into the bacterial suspensions and incubated at 37 °C for 4 h. In parallel, a suspension of *Escherichia coli* (ATCC 25922) was prepared using normal saline at 0.5 Maxwellian turbidity. The freshly-prepared *Escherichia coli* suspension was sprayed on Mueller-Hinton Agar (MHA) solid media within 15 min and dried for 10 min. After that, the sample-treated sterile papers were placed on the *Escherichia coli*-covered culture media and incubated at 37 °C for 24 h. Finally, the diameter of inhibition zones was measured to assess the antibiotic level.

## $\beta$-Lactamase gene profiling with RT-PCR

We first searched the standard sequences of BSBL genes ($bla_{TEM-1}$, $bla_{TEM-2}$, and $bla_{SHV-1}$), ESBL genes ($bla_{TEM}$, $bla_{SHV}$, $bla_{CTX-M}$, $bla_{VEB}$, and $bla_{OXA}$), AmpC genes ($bla_{DHA}$, $bla_{CMY}$, $bla_{FOX}$, $bla_{MOX}$, $bla_{MIR}$, $bla_{ACC}$, $bla_{CIT}$, and $bla_{EBC}$), and carbapenem-resistant genes ($bla_{NDM}$, $bla_{IMP}$, $bla_{KPC}$, $bla_{VIM}$, $bla_{GES}$, $bla_{SME}$, and $bla_{OXA-48}$) from GenBank (https://www.ncbi.nlm.nih.gov/) and designed the primers by Primer 6.0 software. The sequences of the designed primers are shown in Supplementary Tables 6-9. Then, the primers were synthesized by Sangon Biotech, a commercial supplier. The bacterial DNA was extracted by the boiling template method. The 100 diversified body fluid samples were smeared on LB solid medium and cultured for 24 h at 37 °C. Two inoculate ring bacterial lawns were scraped and added into 150 μL PBS (pH 7.8), evenly mixed and boiled for 10 min at 100 °C, then centrifuged at 13800 g for another 10 min. The supernatants containing bacterial DNA were collected and stored at −20 °C. The bacterial DNA and primers were mixed with SGExcel FastSYBR (Thermo Scientific) Mixture consulting the SGExcel FastSYBR qPCR Mixture instructions (Supplementary Table 10). RT-PCR was set according to Supplementary Table 11 and performed by the LightCycler 96 instrument (Roche, Switzerland).

## Statistical analysis

All experiments and assays were repeated at least three times. The data were expressed as mean ± s.d. and compared by two-tailed Student's *t* test. The GraphPad Prism 8 and Origin 8 were used for data analysis.

## Statistics and reproducibility

The SD data are presented in Fig. 2B–E and Supplementary Figs. 32A–D; 34A–F were repeated three times independently with similar results. The SD data presented in Figs. 3C; 4A–F were repeated three times independently. Repeated measurements were analyzed by paired, two-tailed student's *t* test using Graphpad Prism 8 where appropriate, as indicated in each figure legend. No statistical method was used to predetermine sample size and no data were excluded from the analysis.

## Reporting summary

Further information on research design is available in the Nature Portfolio Reporting Summary linked to this article.

## Data availability

All data needed to evaluate the conclusions in the paper are available in the main text or the supplementary information. Source data are provided with this paper. There is no restriction on data availability. No code was developed in this study. Source data are provided with this paper.

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

## Acknowledgements

This work was financially supported in part by funding from the National Key R&D Program of China (2019YFA0210100, D.L.) and the National Natural Science Foundation of China (22174072 and 21977053, D.L.).

## Author contributions
W.L. and D.L. designed the study. W.L. and H.X. performed the experiments. H.G. provided clinical samples and related information. W.L., J.L., and D.L. analyzed the data and wrote the manuscript. D.L. supervised the project. All authors contributed to the manuscript.

## Competing interests
All authors declare no competing interests.
