## [Peer Review file · Nature Communications]

REVIEWER COMMENTS

Reviewer #1 (Remarks to the Author):

Li et al. developed a beta-lactamase subtype visualization chip, enabling rapid and visual identification of clinical isolated resistant bacteria from diverse patient body fluids. This work present promises to enhance the accuracy of antibiotic treatments and contributes to this objective. The concept of the work relies on the colorimetric changes in chromogenic carbapenem substrate and chromogenic cephalosporin substrates. The authors chemically modify the substrates and report the enhancement of the sensitivity of these probes. The modification of the former was recently reported by the group in their paper "Achieving Ultrasensitive Chromogenic Probes for Rapid, Direct Detection of Carbapenemase-Producing Bacteria in Sputum" (JACS-Au 2023). Thus, the extension of this modification route to other well-established substrates and the calibration process (discussed in lines 112 – 244 and includes more than 30 figures in the supporting information) presents a marginal progress. Not to mention that there are numerous works in the literature, including commercially-available assays, that rely on similar concepts. In the next step, the probes are integrated into a simple paper-based sensor (also reported a slightly different format in the JACS-Au paper). Then clinical demonstration of the use of paper sensor for AST of 100 samples is presented and discussed. The sensor presents impressive analytical performance; yet there is no statistical analysis. Thus, I believe that the scientific merit of this work is not at the caliber of Nat. Comm.

Other comments:

- The use of the term "chip" throughout the manuscript (i.e., "this POC testing chip" in the abstract) is confusing as this is scientifically related to the use of silicon technology, and there are AST systems that are indeed silicon chip based such as Li et al., PNAS 2019; etc. This is a paper-based sensor so please use the appropriate term.
- Possibly by presenting the clinical results earlier, the authors can better attract readers and show how their proposed sensor can significantly contribute to more effective antibiotic therapy.
- Figure 3 - please provide statistical comparison to support the limit of detection could be as low as 10⁴ CFU mL⁻¹.
- Figure 6 – Please emphasize the sample 77 and sample 87 in part B which were the under-treated case of the suggested sensor assisted therapy, so the reader can better understand why 2.4 % failed rate in part D.
- Discussion – (1) Please provide the way to deal with the misdiagnosed samples (such as sample 77 and sample 87) in addition to the new design to help with the urgent fungi infections.
- (2) Lines 491 – 492, please provide references when mentioning the accuracy of the antibiotic therapy is less than 50%.

- Methods - Lines 611- 612, it will be better to specify the tested bacterial concentration rather than stating incubation for 1 – 3 hours before testing.

- References – some references appear twice, such as reference 8 and reference 48; reference 31 and reference 46; and reference 38 and reference 49. Reference 51 may be wrongly cited. Is it “CLSI. 2021. Performance Standards for Antimicrobial Susceptibility Testing, M100 31nd Edition. Clinical and Laboratory Standards Institute, Wayne, PA.”?

Reviewer #2 (Remarks to the Author):

In this article, the authors presented a paper chip-based technology for discrimination of β -lactamase subtypes to aid precise antibiotic administration. Firstly, the authors introduced four β -lactam-based chromogenic probes responsive to different β -lactamase subtypes. Their specificity and sensitivity were subsequently examined, showing bathochromic shift, differential response scopes and moderate stability for identification.

Next, the authors constructed chips loaded with probes and reference for visual detection and the smartphone-assisted imaging acquiring and processing were optimized. And detection limits for bacterial strains expressing different β -lactamase subtypes were quantified, showing comparable results.

Lastly, the author applied the chips to detect various real body fluid samples from patients with infection and compare with gold-standard clinical testing method and RT-PCR. The chips demonstrated reliable accuracy for a wide range of samples. And the rapid detection allows intime guidance for precision antibiotics administration.

To the referee, the current manuscript is quite thorough and encouraged to be published in Nature Communication except for some following errors and questions:

1. In the smartphone-based analysis of color changes of chips, the CIELAB (Lab) color model was selected for colorimetry and spectroscopy analysis and a^* values show a close correlation with the positive results. Is there any relationship could be established to indicate the concentration of resistant bacteria and further be applied on how serious the infection could be?

2. The optimum concentration of tartrazine used on the chips was stated to be 0.1mM (line 255) whereas the optimum concentration appears to be around 0.5mM in S31.

3. And also, the detection limits of different resistant bacteria strains is estimated on a order of magnitude but not accurate analytical sensitivity.

4. In Figure 5D, it is hard to distinguish the curves.

5. Figure 1A, the labelling inside figure not clear, can make it more concise and clearer

Reviewer #3 (Remarks to the Author):

The authors have introduced a visually oriented paper chip designed to facilitate the rapid discrimination of β -lactamase subtypes, a critical step in achieving precision antibiotic therapy. Notably, this chip has exhibited remarkable performance in the identification of antibiotic-resistant bacteria, with validation carried out on a substantial dataset of 100 actual patient samples. The results indicate exceptional clinical sensitivity and specificity, both registering at 100%.

Furthermore, in a comparative analysis against empirical medication, the chip has demonstrated the potential to significantly enhance the accuracy of antibiotic selection. Specifically, it improved accuracy from an initial rate of 48% to a substantial 83%. This improvement further escalated to an impressive 97.6% when fungal infection cases were effectively eliminated from consideration.

The utility of the developed β -lactamase subtype visualization (BSV) chip extends beyond the laboratory, as it offers significant promise for bedside or home-based bacterial resistance detection. Consequently, I endorse its publication in Nature Communications. However, there are a few points that merit further clarification and exploration:

1. The driven force of the BSV chip is still not clear, especially before the liquid reaches the paper sheets.

2. The authors should clarify why to select 15 incubation time before the results readout. Additionally, it may be worthwhile to investigate if this incubation time can be shortened without compromising accuracy.

3. The authors used a smartphone and a portable light room to capture the images and read out the results. The device is suitable for the POCT, while the analysis of results seems still labor-tensive. The users need to calculate the quantified results by themselves based on the Lab color model.

4. When using BSV chip to detect bacterial isolates in various body fluid samples, there is a practical problem have to face. That is how to confirm the cultivation time before the usage of BSV chip for different samples of different patients.

5. In figure 5, the authors are suggested to add the culture results in figure 5A and turnaround time in figure 5C.

6. For now, the BSV chip is constructed as a qualification device for the identification of β -lactamase subtypes. Is there possible to develop this BSV chip into a quantification device?

7. Additionally, how about the repeatability of the test performance for the clinical samples? And how long is the shelf-life of BSV chip?

Response to reviewer comments

We are pleased to see the reviewers' enthusiasm toward our manuscript and appreciate their insightful suggestions that have helped strengthen the revised manuscript. Here, we provide a detailed response to each reviewer's comments and the corresponding changes made to the revised manuscript. Please note the following: *Text in italics corresponds to the reviewers' comments*, plain text is our response to the reviewers' comments, and **highlighted text represents the excerpts from the revised Manuscript or Supporting Information.**

Response to Reviewer 1

Li et al. developed a beta-lactamase subtype visualization chip, enabling rapid and visual identification of clinical isolated resistant bacteria from diverse patient body fluids. This work present promises to enhance the accuracy of antibiotic treatments and contributes to this objective. The concept of the work relies on the colorimetric changes in chromogenic carbapenem substrate and chromogenic cephalosporin substrates. The authors chemically modify the substrates and report the enhancement of the sensitivity of these probes. The modification of the former was recently reported by the group in their paper "Achieving Ultrasensitive Chromogenic Probes for Rapid, Direct Detection of Carbapenemase-Producing Bacteria in Sputum" (JACS-Au 2023). Thus, the extension of this modification route to other well-established substrates and the calibration process (discussed in lines 112 – 244 and includes more than 30 figures in the supporting information) presents a marginal progress. Not to mention that there are numerous works in the literature, including commercially-available assays, that rely on similar concepts. In the next step, the probes are integrated into a simple paper-based sensor (also reported a slightly different format in the JACS-Au paper). Then clinical demonstration of the use of paper sensor for AST of 100 samples is presented and discussed. The sensor presents impressive analytical performance; yet there is no statistical analysis. Thus, I believe that the scientific merit of this work is not at the

caliber of Nat. Comm.

Re: First of all, we greatly appreciate the reviewer's invaluable comments and suggestions, which are extremely helpful for us to improve our manuscript.

In the study we published in the journal *JACS Au* (2023, 3, 227-238), we reported an ultrasensitive chromogenic probe that can directly detect carbapenemase-producing bacteria in clinical sputum. We focused on the relationship between the probe structure and enzymatic activity. We found that positively charged substituent (e.g., $-N^+(CH_3)_3$) in the probe could not only boost red-shifting of absorption bands, but also improve molecular interactions with carbapenemases, therefore greatly enhancing the probe sensitivity.

In this present study, we report a POCT diagnostic sensor for rapid and visual identification of β -lactamase subtypes (BSBL, ESBL, AmpC, and carbapenemase) in diverse clinical samples, which significantly enhances the accuracy of antibiotic therapy. With this sensor, the β -lactamase-producing bacteria in 100 real clinical samples were identified within 0.25-3 h, dramatically enhancing antibiotic therapy accuracy from 48% to 83%, and further from 50.6% to 97.6% when fungal infection cases were eliminated from consideration.

Compared to our previous work published in *JACS Au*, this present study focused on the construction of diagnostic platform, achieved the rapid identification of various β -lactamase subtypes, and extended the clinical samples from sputum to a wide range of body fluids including blood, urine, tissue irrigation fluid, and wound drainage fluid. The chromogenic sensor reported in this study was designed for multiple β -lactamase subtypes, providing a new toolbox for guiding precision antibiotic therapy that can not be realized by detecting single β -lactamases alone. This sensor may further help reduce the overuse and dissemination of antibiotics in both hospital and community settings.

Following the reviewer's reminder, we provided the statistical significance to the data in Figure 3B, Figure 5C, and Figure S38, which was determined using an unpaired two-tailed Student's t-test with Welch's correction. Please see the revised **Figure 3B**, **Figure 5C**, and **Figure S38** as below.

Figure 3. Using the BSV sensor to detect the β -lactamase levels in clinically isolated bacteria. (A) Flowchart of the BSV sensor testing process. The bacteria samples were dropped into the sampling hole of the sensor, followed by capturing the image and analyzing the color information with a smartphone. (B) The BSV sensors were used for detecting the clinical isolates of 5 different bacterial lysates at various concentrations from 10^3 to 10^6 CFU/mL. The BSV sensor images were captured after incubation at 15 min. (C) The color changes of the BSV sensor depicted in (B) were quantified using a smartphone equipped with color recognition software. The b^* value and a^* value, representing the transitions from yellow to blue and from green to red respectively, are recorded in the CIELAB (Lab) color model, which is a standardized color space used for colorimetry and spectroscopy analysis in biomedical research. The error bars represent the standard deviations of three replicates.

Figure 5. The BSV sensors were used to identify the β -lactamase subtypes in 100 clinical samples. (A) The BSV testing results were verified by bacterial culture and RT-PCR. (B) The clinical samples involve 13 kinds of body fluids, among which the sputum samples from pneumonia patients rank the top one, reaching 56 cases. Various β -lactamase subtypes were found in these samples. (C) Average \pm s.d. turnaround time of the BSV sensor, bacterial culture, and RT-PCR, which were compared with the AST time in the clinical data (table S12). (D) Accuracy of β -lactamase diagnoses obtained from bacterial culture, BSV sensor, and PT-PCR. The results obtained from the BSV sensor were found to have the same clinical accuracy to the traditional culture-based β -lactamase diagnostic technique and were slightly better than those obtained from RT-PCR.

Figure S38. Statistics of the a* and b* values shown in **Figure S37**.

Comment 1: *The use of the term “chip” throughout the manuscript (i.e., “this POC testing chip” in the abstract) is confusing as this is scientifically related to the use of silicon technology, and there are AST systems that are indeed silicon chip based such as Li et al., PNAS 2019; etc. This is a paper-based sensor so please use the appropriate term.*

Re: We thank the reviewer for the kind suggestion. The term “chip” has been changed into “**sensor**” throughout the revised Manuscript and Supporting Information.

Comment 2: *Possibly by presenting the clinical results earlier, the authors can better attract readers and show how their proposed sensor can significantly contribute to more effective antibiotic therapy.*

Re: We appreciate the reviewer’s constructive suggestion. The clinical results were added in the beginning of the fourth paragraph in the Introduction section, page 3 in the revised Manuscript, “**The β -lactamase-producing bacteria in 100 real clinical samples were discriminated with 100% accuracy within 0.25-3 h, which dramatically enhanced the accuracy of antibiotic therapy from 48% to 83%, and further from 50.6% to 97.6% after eliminating fungal interference.**”

Comment 3: *Figure 3 - please provide statistical comparison to support the limit of detection could be as low as 10^4 CFU mL⁻¹.*

Re: As the reviewer suggested kindly, we supplemented the statistical significance of the data shown in **Figure 3C** using an unpaired two-tailed Student's t-test with Welch's correction. The *P* values < 0.01 indicate a statistical significance, meaning that the lowest detectable concentrations of the BSV sensor were determined to be 10⁵ CFU/mL for *E. coli* (TEM-1) and 10⁴ CFU/mL for *K. pneumoniae* (SHV-18), *M. morganii* (DHA-1), and *E. coli* (NDM-1), respectively. The description was changed in the revised Manuscript, page 3, “This β -lactamase subtype visualization (BSV) sensor could detect clinical bacteria-resistant isolates with the lowest detectable concentration at 10⁴ CFU/mL, 1-2 orders of magnitude lower than other visual β -lactamase probes.³⁵” and page 14, “Therefore, the lowest detectable concentration for *E. coli* (TEM-1) was determined to be 10⁵ CFU/mL (100 CFU), while the others were determined to be 10⁴ CFU/mL (10 CFU) (**Figure 3B**).”

Comment 4: Please emphasize the sample 77 and sample 87 in part B which were the under-treated case of the suggested sensor assisted therapy, so the reader can better understand why 2.4 % failed rate in part D.

Re: Thanks for the reviewer’s kind suggestion. We highlighted the patients NO.77 and NO.87 in **Figure 6B** with yellow background and remarked that they suffered from treatment failures because they were either gram-positive or specific drug-resistant bacterial infection. The corresponding description was added in the revised Manuscript, pages 21-22, “Therefore, we excluded the fungal infections from the 100 testing cases (indicated by the red boxes in **Figure 6B**), after which the accuracy of antibiotic therapy was enhanced from 50.6% empirically to an astounding 97.6%, yet there were still 2 cases misdiagnosed (patients NO.77 and NO.87 labeled with yellow background).” Please see the revised **Figure 6** below.

Figure 6. The BSV sensor-assisted precision antibiotic therapy that was compared with empirical medication. (A) A list of medications classified by β -lactamase subtypes. Note that the cases in the absence of β -lactamases can be referred to the Sanford Guide to Antimicrobial Therapy (2022). (B) The prescriptions that were made empirically and the same ones were made according to the BSV sensor results. (C) The BSV sensor can eliminate antibiotic overuse, reduce the proportion of under-treated cases from 41% to 17%, and enhance the correct treatment from 43% to 83%. (D) By excluding fungal interference, the BSV sensor could further reduce the fraction of under-treated cases from 30.6% to 2.4% while increasing the correct treatment from 50.6% to 97.6%.

Comment 5: Please provide the way to deal with the misdiagnosed samples (such as sample 77 and sample 87) in addition to the new design to help with the urgent fungi infections.

Re: Thanks for the kind reminder from the reviewer. The corresponding description was added in the revised Manuscript, page 23, “Urgent fungal testing can only be performed using RT-PCR or test kits that correlate with the source of the infection; otherwise, the physician must wait for the results of the microbiological identification from the laboratory before administering treatment.”

Comment 6: Lines 491 – 492, please provide references when mentioning the accuracy of the antibiotic therapy is less than 50%.

Re: Thanks for the kind reminder from the reviewer. The information “Of all 18848864 antibiotic prescriptions, 9689937 (51.4%) were inappropriate, 5354224 (28.4%) were potentially appropriate, 2893102 (15.3%) were appropriate, and 911601 (4.8%) could not be linked to any diagnosis.” is described in Reference 10, which has been cited in the revised Manuscript, page 22, “At present, primary antibiotic therapy mainly depends on the physicians’ empirical assessment of the infection situation and medication knowledge, by which the accuracy of antibiotic therapy is overall less than 50%.¹⁰”

Comment 7: Methods - Lines 611- 612, it will be better to specify the tested bacterial concentration rather than stating incubation for 1 – 3 hours before testing.

Re: Thanks for the kind reminder from the reviewer. The bacterial concentrations in most clinical samples are frequently below the detectable concentration (10^4 CFU/mL). Therefore, these samples require bacterial culture for different time to increase the bacterial concentrations above the detection sensitivity of the sensor. As shown in **Figure 4**, for example, the antibiotic-resistant bacterial isolates (2853 CFU/mL) in a pus sample required 1 h of cultivation to increase the counts to 8716 CFU/mL for testing. For lavage fluid, drainage fluid, and urine samples, they needed 2 h of cultivation to increase their counts from 2020, 1863, and 1756 CFU/mL to 13540, 13750, and 7580 CFU/mL, respectively. With respect to the blood sample, its antibiotic-resistant bacterial count was extremely low (723 CFU/mL), which required a longer time (3 h) of cultivation to increase its count to 8536 CFU/mL for visual detection. This

information was added in the revised Manuscript, page 27, “Except for sputum samples that could be directly tested, the other samples were cultured for 1-3 h to increase the bacterial concentrations above the detection sensitivity of the sensor before testing.”

Comment 8: References – some references appear twice, such as reference 8 and reference 48; reference 31 and reference 46; and reference 38 and reference 49. Reference 51 may be wrongly cited. Is it “CLSI. 2021. Performance Standards for Antimicrobial Susceptibility Testing, M100 31nd Edition. Clinical and Laboratory Standards Institute, Wayne, PA.”?

Re: We greatly appreciate the kind reminder from the reviewer. The mistakes about the references have been corrected throughout the revised manuscript.

We greatly appreciate the reviewer’s insightful comments and helpful suggestion again.

Response to Reviewer 2

In this article, the authors presented a paper chip-based technology for discrimination of β -lactamase subtypes to aid precise antibiotic administration. Firstly, the authors introduced four β -lactam-based chromogenic probes responsive to different β -lactamase subtypes. Their specificity and sensitivity were subsequently examined, showing bathochromic shift, differential response scopes and moderate stability for identification.

Next, the authors constructed chips loaded with probes and reference for visual detection and the smartphone-assisted imaging acquiring and processing were optimized. And detection limits for bacterial strains expressing different β -lactamase subtypes were quantified, showing comparable results.

Lastly, the author applied the chips to detect various real body fluid samples from patients with infection and compare with gold-standard clinical testing method and RT-PCR. The chips demonstrated reliable accuracy for a wide range of samples. And the rapid detection allows intime guidance for precision antibiotics administration.

To the referee, the current manuscript is quite thorough and encouraged to be published in Nature Communication except for some following errors and questions.

Re: We thank the reviewer for the supportive comments and helpful suggestions.

Comment 1: *In the smartphone-based analysis of color changes of chips, the CIELAB (Lab) color model was selected for colorimetry and spectroscopy analysis and a^* values show a close correlation with the positive results. Is there any relationship could be established to indicate the concentration of resistant bacteria and further be applied on how serious the infection could be?*

Re: We appreciate the valuable suggestion. During our experiments, we attempted to establish a relationship between the a^* value and the concentration of drug-resistant bacteria, but we discovered that the enzyme activity varies with the isoforms. For example, both NDM-1 and OXA-48 belong to carbapenemase, but NDM-1 shows much higher hydrolysis activity than the same concentration of OXA-48 towards CCS-

$N^+(CH_3)_3$ (**Figure 2**). Hence, the a^* value can only indicate the drug-resistant bacteria's capacity to hydrolyze the probes (antibiotic analogues), while the concentration of the drug-resistant bacteria cannot be determined accurately.

Comment 2: The optimum concentration of tartrazine used on the chips was stated to be 0.1 mM (line 255) whereas the optimum concentration appears to be around 0.5 mM in S31.

Re: Tartrazine was used as a negative control, where the b^* values of the samples should be kept consistent before and after the test. So it seemed that the optimal concentrations were 0.1 mM or 0.3 mM due to their minimal variation before and after the test (**Figure S31A** in the revised Supporting Information). However, although the color at 0.5 mM are intense, but the b^* values showed remarkable difference before and after the test. Therefore, we chose 0.1 mM tartrazine.

Comment 3: And also, the detection limits of different resistant bacteria strains is estimated on a order of magnitude but not accurate analytical sensitivity.

Re: We appreciate the valuable suggestion. The detection limits were inappropriately used in our initially submitted manuscript because detection limit (or limit of detection, LOD) is defined as 3-fold standard deviation of the regression line /slope(S) from regression line. In our paper, we tested the samples with gradually diluted concentrations of bacteria other than estimating their detection limits with a regression line. Therefore, we replaced “detection limits” with “lowest detectable concentrations” in the revised Manuscript, page 3, “This β -lactamase subtype visualization (BSV) sensor could detect clinical bacteria-resistant isolates with the lowest detectable concentration at 10^4 CFU/mL, 1-2 orders of magnitude lower than other visual β -lactamase probes.”, page 14, “Therefore, the lowest detectable concentration for *E. coli* (TEM-1) was determined to be 10^5 CFU/mL (100 CFU), while the others were determined to be 10^4 CFU/mL (10 CFU) (**Figure 3B**).”

Comment 4: In Figure 5D, it is hard to distinguish the curves.

Re: We appreciate the kind reminder! We opted to exhibit the histograms of bacterial culture, PT-PCR, and BSV sensor to highlight the detection accuracy of β -lactamase subtypes clearly. Please see the revised **Figure 5** below.

Figure 5. The BSV sensors were used to identify the β -lactamase subtypes in 100 clinical samples. (A) The BSV testing results were verified by bacterial culture and RT-PCR. (B) The clinical samples involve 13 kinds of body fluids, among which the sputum samples from pneumonia patients rank the top one, reaching 56 cases. Various β -lactamase subtypes were found in these samples. (C) Average \pm s.d. turnaround time of the BSV sensor, bacterial culture, and RT-PCR, which were compared with the AST time in the clinical data (table S12). (D) Accuracy of β -lactamase diagnosis

obtained from bacterial culture, BSV sensor, and PT-PCR. The results obtained from the BSV sensor were found to have the same clinical accuracy to the traditional culture-based β -lactamase diagnostic technique and were slightly better than those obtained from RT-PCR.

Comment 5: Figure 1A, the labelling inside figure not clear, can make it more concise and clearer.

Re: We appreciate the kind reminder! The labeling in **Figure 1A** was improved to be clearer than the initially submitted version. Please see the revised **Figure 1** below.

Figure 1. Design of the BSV sensor for rapid detection of antibiotic resistance. (A) The management of antibiotic resistance by conventional strategies and the BSV sensor. In most countries, healthcare institutions are commonly categorized into primary (no AST), secondary, and tertiary levels. Typically, suspected infection cases are initially managed to control symptoms in clinical settings

based on the physician's experience, and the medication would be adjusted according to the bacterial culturing results several days later. By contrast, the BSV **sensor** can report the β -lactamase subtypes of various body fluid samples from suspected infection patients within 0.25-3 h. (B) Design of the BSV **sensor** and working principle of the 6 independent sample chambers. Chamber 1 is loaded with tartrazine to reflect the quality of the BSV **sensor**, while chamber 2 makes use of bromophenol blue to indicate the presence of proteins. Chambers 3 to 6 are loaded with the four as-synthesized β -lactamase chromogenic probes to respectively differentiate BSBL, ESBL, AmpC, and carbapenemases. (C) Hydrolysis properties of the four β -lactamase subtypes towards the chromogenic probes.

Response to Reviewer 3

The authors have introduced a visually oriented paper chip designed to facilitate the rapid discrimination of β -lactamase subtypes, a critical step in achieving precision antibiotic therapy. Notably, this chip has exhibited remarkable performance in the identification of antibiotic-resistant bacteria, with validation carried out on a substantial dataset of 100 actual patient samples. The results indicate exceptional clinical sensitivity and specificity, both registering at 100%.

Furthermore, in a comparative analysis against empirical medication, the chip has demonstrated the potential to significantly enhance the accuracy of antibiotic selection. Specifically, it improved accuracy from an initial rate of 48% to a substantial 83%. This improvement further escalated to an impressive 97.6% when fungal infection cases were effectively eliminated from consideration.

The utility of the developed β -lactamase subtype visualization (BSV) chip extends beyond the laboratory, as it offers significant promise for bedside or home-based bacterial resistance detection. Consequently, I endorse its publication in Nature Communications. However, there are a few points that merit further clarification and exploration.

Re: We thank the reviewer for the supportive comments and helpful suggestions.

Comment 1: *The driven force of the BSV chip is still not clear, especially before the liquid reaches the paper sheets.*

Re: We appreciate the kind reminder! The relevant information was added in the revised Manuscript, page 4, “The samples (20 μ L) are dropped into the central sampling hole and then quickly diffuse into the chambers via capillary siphoning.” and in the revised Supporting Information, page S24, “A 2 mm diameter sized hole in the bottom of the chambers is linked to the central sampling hole by a channel (6 mm length \times 1 mm width), where the sample can be syphoned into the chambers in a few seconds.”

Comment 2: *The authors should clarify why to select 15 incubation time before the*

results readout. Additionally, it may be worthwhile to investigate if this incubation time can be shortened without compromising accuracy.

Re: We appreciate the valuable suggestion. We added this information in “**Visual detection of the clinical isolates**” and “**Rapid POC detection of diverse body fluid samples**” in the **Methods** part of the article. A further 10 minutes of bacterial lysis with Triton was conducted before testing the samples, followed by 5 minutes of color development on the paper. We fixed the whole assay time as 15 minutes for the sake of experimental rigor.

Comment 3: *The authors used a smartphone and a portable light room to capture the images and read out the results. The device is suitable for the POCT, while the analysis of results seems still labor-tensive. The users need to calculate the quantified results by themselves based on the Lab color model.*

Re: We appreciate the kind reminder! We took pictures of the BSV sensor with a smartphone under a shadowless lamp. The pictures were then dealt with a color recognizer software to convert the colors to the Lab color model, finally yielding the quantifiable a^* values. As illustrated in the picture below, smartphones may instantly obtain 8 color model results by using the **color picker** APP. If we take the photograph without a shadowless lamp, we simply need to change the brightness of the photograph on the phone before color reading. So, no matter at hospital, community, or home, the testing is simple to conduct.

Comment 4: *When using BSV chip to detect bacterial isolates in various body fluid samples, there is a practical problem have to face. That is how to confirm the cultivation time before the usage of BSV chip for different samples of different patients.*

Re: Thanks for your concerns. The bacterial concentrations in most clinical samples are frequently below the detectable concentration (10^4 CFU/mL). Therefore, these samples require bacterial culture for different time to increase the bacterial concentrations above the detection sensitivity of the sensor. As shown in **Figure 4**, we counted the bacteria in six different types of samples at various stages of incubation. To reach the detectable concentrations, blood samples should be incubated for 3 hours; lavage, drainage, and urine samples should be incubated for 2 hours; pus samples should be incubated for 1 hour; and no culture is required for sputum samples. 100 clinical samples were tested for assessing the antibiotic resistance using the above-mentioned technique. The results demonstrate that various sorts of samples could be reliably identified after specific incubation time we optimized in this study.

Comment 5: *In figure 5, the authors are suggested to add the culture results in figure 5A and turnaround time in figure 5C.*

Re: We appreciate the kind reminder! Please see the revised **Figure 5** below.

Figure 5. The BSV sensors were used to identify the β -lactamase subtypes in 100 clinical samples. (A) The BSV testing results were verified by bacterial culture and RT-PCR. (B) The clinical samples involve 13 kinds of body fluids, among which the sputum samples from pneumonia patients rank the top one, reaching 56 cases. Various β -lactamase subtypes were found in these samples. (C) Average \pm s.d. turnaround time of the BSV sensor, bacterial culture, and RT-PCR, which were compared with the AST time in the clinical data (table S12). (D) Accuracy of β -lactamase diagnosis obtained from bacterial culture, BSV sensor, and PT-PCR. The results obtained from the BSV sensor were found to have the same clinical accuracy to the traditional culture-based β -lactamase diagnostic technique and were slightly better than those obtained from RT-PCR.

Comment 6: *For now, the BSV chip is constructed as a qualification device for the identification of β -lactamase subtypes. Is there possible to develop this BSV chip into a quantification device?*

Re: Thanks for your question. In our experiments, we attempted to establish a quantitative relationship between the a^* value and the concentration of the drug-resistant bacteria, but we discovered that the enzyme activity varies with the isoforms. For example, both NDM-1 and OXA-48 belong to carbapenemase, but NDM-1 shows much higher hydrolysis activity than the same concentration of OXA-48 towards CCS- $N^+(CH_3)_3$ (**Figure 2**). Hence, the a^* value can only indicate the drug-resistant bacteria's capacity to hydrolyze the probes (antibiotic analogues), while the concentration of the drug-resistant bacteria cannot be determined accurately.

Comment 7: *Additionally, how about the repeatability of the test performance for the clinical samples? And how long is the shelf-life of BSV chip?*

Re: Thanks for your question. The results in **Figure 2**, **Figure 3**, and **Figure 4** for detecting drug-resistant bacteria were all replicated three times with high consistency and accuracy. The four probes' solid powder can be kept in a dark environment at 50° C for 45 days without deterioration, and the chip can be kept in a dark environment at 4° C for more than 6 months without deterioration.

REVIEWERS' COMMENTS

Reviewer #2 (Remarks to the Author):

In this revised manuscript, the authors addressed the referees' queries, and mainly highlighted their focus on the construction of diagnostic platform for the rapid identification of various beta-lactamase subtypes and further extension their testing in the clinical samples from sputum to a wide range of body fluids. The authors also cleared the technical concerns by further testing and discussion in the main context. The referee concluded that the revised manuscript has been largely improved, and it can be considered for publication in Nature Communications.

Reviewer #3 (Remarks to the Author):

The authors have satisfactorily addressed all of my comments. I recommend accepting the manuscript at this time.